# Towards Execution-Grounded Automated AI Research

Chenglei Si [* 1]  Zitong Yang [* 1]  Yejin Choi [1]  Emmanuel Candès [1]  Diyi Yang [1]  Tatsunori Hashimoto [1]

## Abstract

Automated AI research holds great potential to accelerate scientific discovery. However, current LLMs often generate plausible-looking but ineffective ideas. Execution grounding may help, but it is unclear whether automated execution is feasible and whether LLMs can learn from the execution feedback. To investigate these, we first build an automated executor to implement ideas and launch large-scale parallel GPU experiments to verify their effectiveness. We then convert two realistic research problems – LLM pre-training and post-training – into execution environments and demonstrate that our automated executor can implement a large fraction of the ideas sampled from frontier LLMs. We analyze two methods to learn from the execution feedback: evolutionary search and reinforcement learning. Execution-guided evolutionary search is sample-efficient: it finds a method that significantly outperforms the GRPO baseline on post-training, and finds a pre-training recipe that outperforms the nanoGPT baseline on pre-training, all within just ten search epochs. Frontier LLMs often generate meaningful algorithmic ideas during search, but they tend to saturate early and only occasionally exhibit scaling trends. Reinforcement learning from execution reward, on the other hand, suffers from mode collapse. It successfully improves the average reward of the ideator model but not the upper-bound, due to models converging on simple ideas. We thoroughly analyze the executed ideas and training dynamics to facilitate future efforts.

## 1. Introduction

We envision automated AI research: LLMs generate research ideas to tackle important research problems, imple-

---
[*]Equal contribution  [1]Stanford University.  Correspondence to: Chenglei Si <clsi@stanford.edu>, Zitong Yang <zitong@berkeley.edu>.

*Proceedings of the 43rd International Conference on Machine Learning*, Seoul, South Korea. PMLR 306, 2026. Copyright 2026 by the author(s).

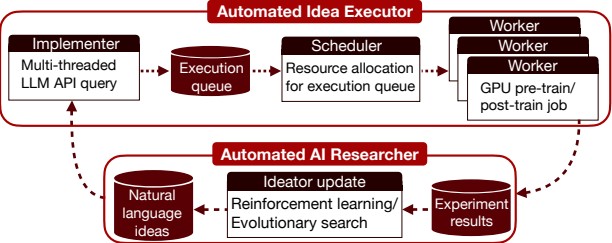

*Figure 1.* We build an automated idea executor involving Implementer, Scheduler, and Worker. We then use this automated executor as a reward function to teach LLMs to generate more effective ideas through evolutionary search and RL. We only update the ideator in the learning process.

ment the ideas as code, run experiments to verify the effectiveness, and continuously learn from the execution results. If successful, these automated AI researchers could automatically develop and identify effective research ideas in a massive search space, thereby scalably converting compute into scientific discovery. Despite the promise, automated AI research is bottlenecked by the ability of LLMs to generate effective ideas. Si et al. (2025b) and Si et al. (2025a) evaluated the quality of LLM-generated research ideas through large-scale expert review and found that LLM ideas often look convincing but are ineffective after being executed by human researchers.

This highlights the need to ground idea generation in execution. However, obtaining execution results of ideas in an automated and scalable manner is challenging, especially since we are targeting open-ended AI research where any ideas expressible in natural language are within our action space. To tackle this, we design and build a high-throughput automated idea executor that can implement hundreds of model-generated ideas and execute them in parallel to obtain the experiment results as execution feedback.

To study the extent to which we can automate realistic LLM research, we chose two GPU-intensive research problems (LLM pre-training and post-training) that are critical for improving the capabilities of LLMs as the research environments for our automated AI researchers. For the first time, we demonstrate that our automated executor can implement a large fraction of LLM-generated ideas on these challenging open-ended research problems, with over 90% execution rates on the pre-training environment with Claude-

4.5-Sonnet and Claude-4.5-Opus.

To analyze whether grounding on execution-feedback can improve LLM idea generation, we define objective performance metrics for both environments and analyze the strengths and weaknesses of two popular learning algorithms: evolutionary search and reinforcement learning.

We use our automated executor to guide evolutionary search. Within ten search epochs, this execution-guided search finds a post-training recipe that outperforms the GRPO baseline (69.4% vs 48.0%) on the task of post-training a 1.5B model for math reasoning, and a pre-training recipe that outperforms the nanoGPT baseline (19.7 minutes vs 35.9 minutes) on the task of minimizing the training wall-clock time to reach the target validation loss (Table 1). Our analysis shows that models are often generating algorithmic ideas apart from tuning hyper-parameters, and evolutionary search significantly outperforms best-of-N under the same sampling budget. However, when analyzing the scaling trend, only Claude-4.5-Opus shows a clear scaling curve, while both Claude-4.5-Sonnet and GPT-5 tend to saturate early.

We then use the automated executor as the reward function in an RL loop to finetune Qwen3-30B. We show that RL with execution reward can successfully improve the average reward of the ideator model, similar to typical RL from verifiable rewards. However, RL does not improve the max reward, which is the more important metric for scientific discovery. In fact, we reveal that RL causes the ideator model to converge on a few easy-to-implement ideas, resulting in a collapse in thinking length and idea diversity.

In summary, we develop a large-scale automated idea executor system that can implement research ideas for open-ended and realistic research problems. Using this automated executor, we conduct an in-depth analysis of how well LLM ideators can learn from execution feedback to improve effectiveness through evolutionary search and RL. Execution-guided evolutionary search is sample-efficient and effective, but shows limited scaling. RL from execution reward suffers from diversity collapse and does not improve the upper-bound. We additionally provide extensive analysis on the executed ideas and suggest promising directions to improve the existing learning algorithms. Altogether, we demonstrate the feasibility and potential of grounding LLM ideation in automated execution and uncover important limitations for future improvement.

## 2. Automated Idea Executor

To measure the effectiveness of model-generated ideas, we build an automated executor that takes natural language research ideas as input, generates code implementations, runs the experiments on the backend, and returns the idea's benchmark performance as the final output.

*Table 1.* Performance of our execution-guided search in comparison with the provided baselines and best human experts. Metrics are validation accuracy and validation loss.

| | Post-Training$_\uparrow$ | Pre-Training$_\downarrow$ |
|---|---|---|
| Baseline | 48.0% | 35.9 min |
| Execution-Guided Search | 69.4% | 19.7 min |
| Best Human Expert | 68.8% | 2.1 min |

### 2.1. Research Environments for Ideation

Our automated idea executor is grounded in specific research environments, where each environment consists of a research problem, a baseline codebase, a benchmark to measure performance on, fixed training and evaluation data, and evaluation metrics. When constructing the research environments, we aim to select research problems that are open-ended, so that there is ample room for new algorithmic innovations, and at the same time have well-established baselines and benchmarking metrics so that measuring effectiveness is straightforward. In this work, we construct both a pre-training environment and a post-training environment for the automated AI researchers to work on.

**Pre-Training Task: Improving nanoGPT** In the nanoGPT environment, we provide a baseline codebase adapted from the nanoGPT speedrun (Jordan et al., 2024) and ask the ideator model to brainstorm possible improvements. The original speedrun task is to minimize the time to pre-train a 124M GPT-2 model (Radford et al., 2019) on the FineWeb corpus (Penedo et al., 2024) to reach a validation loss of 3.28 on the validation set. We did several modifications to the original speedrun setting. First, we introduce a proxy reward equal to the reciprocal of the validation loss ($\frac{1}{loss}$) when performing the search and RL experiments in later sections of the paper. This way, we can fix the training wall-clock time to be 25 minutes and ask the model to directly optimize the proxy reward under this fixed budget, so that we can avoid different runs having vastly different runtimes. We report the validation loss or the proxy reward metric in most plots, and only measure and report the training time metric for the top solution in order to directly compare it with the human experts' solutions on the original nanoGPT speedrun leaderboard. Second, to avoid any possible reward hacking, we freeze all evaluation hyper-parameters and implement an inference function that predicts one future token at a time to prevent models from changing the attention mechanism in a way that leaks future tokens (which happened multiple times during our initial development). We use this inference function during the final validation after each training run.

**Post-Training Task: Improving GRPO** In the GRPO environment, the baseline is an implementation of the GRPO algorithm (Shao et al., 2024) that finetunes a Qwen2.5-Math-1.5B checkpoint (Yang et al., 2024) on the MATH

dataset (Hendrycks et al., 2021). The ideator model needs to brainstorm post-training algorithms more effective than the baseline. We specify a fixed training wall-clock time budget and use the max accuracy on the MATH validation set during training as the metric. To prevent reward hacking, we keep all validation-related code in a separate file and do not allow the automatic executor to access or modify it.

## 2.2. System design

The automated idea executor can be viewed as a high-level API whose input is a batch of natural language ideas, and the output is the benchmark performance of each idea. There are three core building blocks of this API (Figure 1): *Implementer* – the server that generates the code diff for the idea and applies those changes; *Scheduler* – a middle layer that receives the list of codebases and allocates resources to run experiments; *Worker* – the cluster with GPU available that runs the experiments and uploads the experiment results.

**Implementer** The implementer is hosted on a CPU machine with high IO capacity. First, the user submits a batch of natural language ideas. Then, for each idea, the implementer makes parallelized API calls to the code execution LLM to obtain a `diff` file that can be patched into the corresponding baseline codebase. To optimize for efficiency, we prompt the code execution LLM with both the idea and the baseline codebase to sample 10 code diff files in parallel. For each sample, if the generated diff file cannot be patched into the original codebase, we provide the patch log and ask the model to revise the original generation. We repeat this sequential self-revision for a maximum of 2 times. In the end, we return the first code diff file that can be successfully patched into the baseline codebase. The patched codebase is then submitted to a cloud bucket as a `.zip` file.

**Scheduler** Under a set clock frequency, the scheduler downloads the new codebases from the cloud. If the codebase has not been executed, the scheduler examines the resource requirement of the given research environment and prepares a job configuration to be submitted.

**Worker** Once the scheduler finds available resources, it connects the prepared job configuration with the GPU resource and initializes the worker to run the experiment. If the execution of the experiment is successful, the worker will upload the experiment logs including all performance metrics to another cloud bucket (`wandb`) along with the complete metadata: idea content, code change, execution log, etc. If the execution fails (e.g., due to bugs in code implementation), the worker halts. The user (i.e., the ideator model) can then download the execution results and see the performance of the batch of ideas they submitted with full training logs.

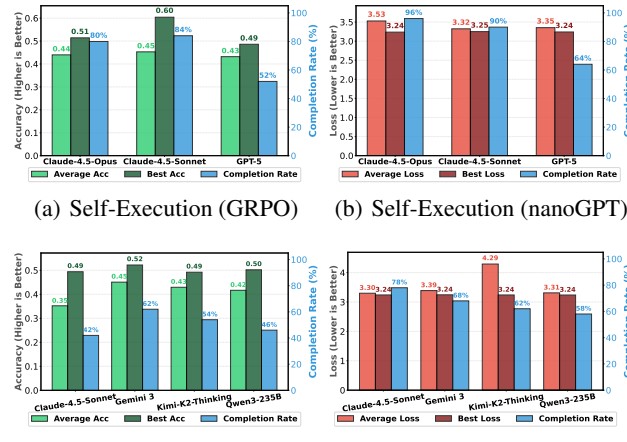

(a) Self-Execution (GRPO)  (b) Self-Execution (nanoGPT)

(c) GPT-5 Execution (GRPO)  (d) GPT-5 Execution (nanoGPT)

*Figure 2.* Model performance comparison with self-execution (top row) vs GPT-5 execution (bottom row) on the GRPO and nanoGPT environments. The baseline accuracy for GRPO is 0.480, and the baseline loss for nanoGPT is 3.255. The completion rate is high for most models, especially under self-execution.

## 3. Benchmarking LLM Ideators and Executors

The prerequisite for an execution-grounded feedback loop is that current LLMs can serve as both ideators and executors, so that we can get meaningful reward signals for the models to learn from. To examine this prerequisite, we first benchmark various frontier LLMs as both the ideator and the executor.

### 3.1. End-to-End Ideation and Execution

In the first setting, we sample ideas from an LLM, and use the same LLM as the code execution model to execute its own ideas. We sample and execute 50 ideas from Claude-4.5-Opus, Claude-4.5-Sonnet, and GPT-5, and measure several metrics: (1) completion rate: the percentage of ideas that are successfully executed with a valid (non-zero) experiment result after execution; (2) average performance: the average validation accuracy or loss for all the successfully executed ideas among the 50 samples; (3) best performance: the highest validation accuracy or lowest validation loss among all executed ideas. We present results in the top row of Figure 2. Notably, a large fraction of the sampled ideas can indeed be executed successfully, with Claude-4.5-Opus and Claude-4.5-Sonnet having a significantly higher execution rate than GPT-5. Moreover, the best-of-N performance ($N = 50$) of these models can already beat the original baseline solutions. For example, on the GRPO environment, Claude-4.5-Sonnet gets a max accuracy of 60.4% as compared to the baseline of 48.0%; on nanoGPT, Claude-4.5-Opus gets a lowest loss of 3.237 as compared to the baseline of 3.255.

---

**Algorithm 1** Execution-Guided Search

---

**Require:** batch size $N$, epochs $T$, baseline performance $\beta$
**Require:** initial exploitation rate $a_1 \in [0, 100]$, annealing schedule $a(t)$ for $t \in \{1, \dots, T\}$
1: Sample initial batch of ideas $\mathcal{I}_0 \leftarrow \text{SAMPLEIDEAS}(N)$
2: Execute $\mathcal{I}_0$ to obtain trajectories $\mathcal{D}_0 \leftarrow \{(\text{idea}, \text{reward})\}$
3: **for** $t = 1$ **to** $T$ **do**
4:     $a \leftarrow a(t)$             $\triangleright (100 - a)\%$ exploration rate
5:     $\mathcal{D}_{<t} \leftarrow \bigcup_{k=0}^{t-1} \mathcal{D}_k$
6:     $\mathcal{D}^+ \leftarrow \{(i, r) \in \mathcal{D}_{<t} \ : \ r > \beta\}$   $\triangleright$ positive trajectories
7:     $N_{\text{exp}} \leftarrow \lfloor \frac{a}{100} N \rfloor, \quad N_{\text{expl}} \leftarrow N - N_{\text{exp}}$
8:     $\mathcal{I}_t^{\text{exp}} \leftarrow \text{EXPLOITVARIANTS}(\mathcal{D}^+, N_{\text{exp}})$
9:     $\tilde{\mathcal{D}}_{<t} \leftarrow \text{SUBSAMPLETOCONTEXT}(\mathcal{D}_{<t})$
10:    $\mathcal{I}_t^{\text{expl}} \leftarrow \text{EXPLORENOVEL}(\tilde{\mathcal{D}}_{<t}, N_{\text{expl}})$
11:    $\mathcal{I}_t \leftarrow \mathcal{I}_t^{\text{exp}} \cup \mathcal{I}_t^{\text{expl}}$
12:    Execute $\mathcal{I}_t$ to obtain trajectories $\mathcal{D}_t \leftarrow \{(\text{idea}, \text{reward})\}$
13: **end for**
14: **return** $\bigcup_{t=0}^{T} \mathcal{D}_t$

---

### 3.2. Comparing Ideators with the Same Executor

In the second setting, we fix the executor model to be GPT-5 and use different ideator models to sample ideas. As shown in the bottom row of Figure 2, even when the ideator and executor are different models, the execution rate is still decent (ranging from 42% to 78%), although we do notice that the same ideas from Claude-4.5-Sonnet get a lower execution rate when executed by GPT-5 instead of itself (84% vs 42% on GRPO and 90% vs 78% on nanoGPT). Moreover, frontier open-weight models like Kimi-K2-Thinking (Kimi Team, 2025) and Qwen3-235B-A22B (Yang et al., 2025a) can also get non-trivial completion rates and achieve best-of-N performance that outperforms the baseline solutions in this setting. For example, Qwen3-235B achieves a max accuracy of 50.2% on GRPO and min loss of 3.238 on nanoGPT with $N = 50$, both better than the baselines.

These benchmarking results demonstrate the feasibility of the automated ideation and execution loop. Next, we build search scaffolds and RL training loops to examine whether models can learn from the execution feedback.

## 4. Execution-Guided Evolutionary Search

We develop an evolutionary search scaffold on top of frontier LLMs to optimize for effective ideas based on execution feedback. We introduce our search method that blends exploration and exploitation, its effectiveness on our two research environments, and various analyses of the generated ideas throughout the evolutionary search process.

### 4.1. Search Scaffold

Our search method is inspired by prior evolutionary search approaches for code optimization, such as AlphaEvolve (Novikov et al., 2025). Our algorithm is detailed in

Algorithm 1. At the first search epoch, we sample a full batch of new ideas. In all subsequent epochs, we split the idea generation into exploitation and exploration subsets. For exploitation, we choose ideas from previous epochs that outperform the baseline and append them to the idea generation prompt to ask the ideator model to generate new variants that combine their strengths. For exploration, we randomly sample ideas from previous epochs to append to the idea generation prompt until reaching the max context length and instruct the ideator model to generate completely new ideas different from them. We start with 50% exploitation and 50% exploration at epoch 1 and gradually anneal the exploration rate and increase the exploitation ratio in later epochs. We use a batch size of 50 for the GRPO environment and a batch size of 80 for the nanoGPT environment.

### 4.2. Experiment Results

For each environment, we perform execution-guided search with three different models: Claude-4.5-Opus, Claude-4.5-Sonnet, and GPT-5. For each experiment, we use the same model as both the ideator and executor (i.e., self-execution). We plot the progression of the best performance at each search epoch in Figure 3. We summarize several notable trends below.

First, we observe a scaling trend with Claude-4.5-Opus, where searching for more epochs leads to a higher upper bound. In contrast, Claude-4.5-Sonnet and GPT-5 tend to saturate early. Second, all models can find ideas that significantly outperform the baselines. On GRPO, Claude-4.5-Sonnet finds that using vanilla policy gradient with the group-average baseline without importance reweighting or clipping outperforms the standard GRPO objective in this particular experiment setup and exploits this finding in all subsequent search epochs, resulting in the best solution of 69.4% at epoch 2 with precise hyper-parameter tuning. On nanoGPT, Claude-4.5-Opus achieves the min validation loss of 3.1407 at epoch 9 by combining various architectural modifications, hyper-parameter tuning, and applying exponential moving average of intermediate checkpoints during validation (see Appendix A.2 for the full idea). We run this top solution on 8 H100s to follow the same setup as the nanoGPT speedrun, and it reaches the 3.28 target validation loss in 19.7 minutes, a significant speedup as compared to the baseline codebase, which takes 35.9 minutes of training time to reach the same target validation loss.

To better contextualize these solutions optimized by the model, we also compare the top performance of our execution-guided search to human experts (Table 1). For the GRPO environment, we compare with the leaderboard of a graduate-level LLM class, which hosted the same environment as an assignment for all students to optimize the validation accuracy under the same training time budget. The

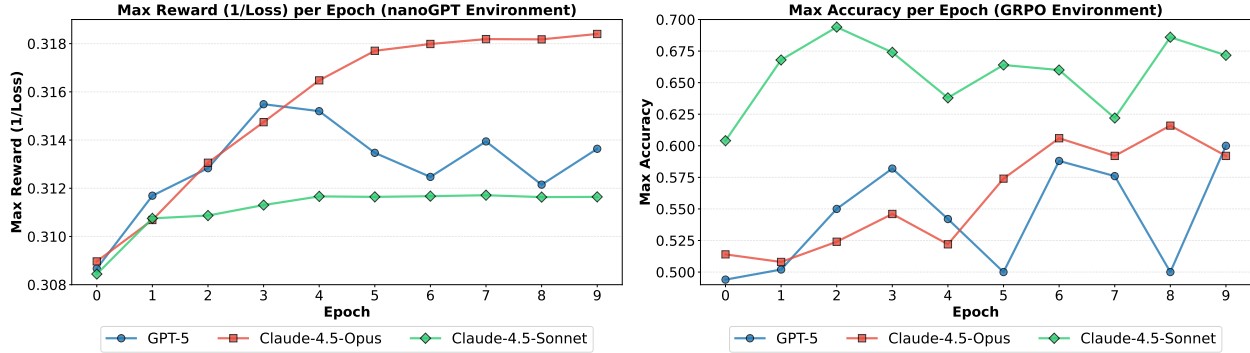

*Figure 3.* Best performance at each epoch when performing execution-guided search with different models. For the nanoGPT environment (left), we use the reciprocal of the validation loss as the metric; for the GRPO environment (right), we use validation accuracy as the metric. Claude-4.5-Opus exhibits a scaling trend on both environments and achieves the best performance on nanoGPT. Claude-4.5-Sonnet achieves the best performance on GRPO due to effective hyper-parameter tuning, but saturates early.

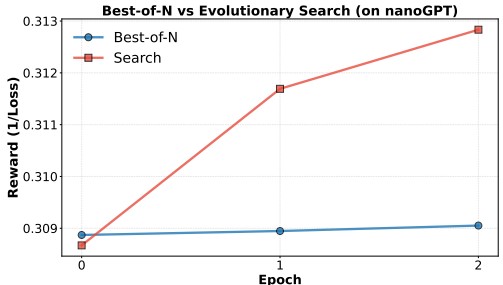

*Figure 4.* Comparison between best-of-N (blue) and our execution-guided search (red) under the same sampling budget.

best student solution achieved an accuracy of 68.8%, lower than Claude-4.5-Sonnet's top solution using our execution-guided search. For the nanoGPT environment, we directly compare with the nanoGPT speedrun leaderboard, [1] where the top human solution as of December 2025 can reach the target validation loss under 2.1 minutes, indicating significant headroom for further model capability and search method improvement on this environment.

### 4.3. Comparison with Best-of-N

To demonstrate the effectiveness of our search scaffold, we compare our execution-guided search with the best-of-N baseline with the same sampling budget on the nanoGPT environment. Since the batch size for our search is 80, we compare the first 3 epochs of the execution-guided search using the GPT-5 backbone with the best-of-N results of GPT-5 with $N \in \{80, 160, 240\}$. As shown in Figure 4, search and best-of-N start from similar performance at epoch 0 (they are not exactly the same due to variances from sampling), but evolutionary search significantly outperforms best-of-N from epoch 1 onward, demonstrating that the model is

effectively leveraging trajectories from previous epochs to generate more effective ideas in future epochs.

### 4.4. Analysis of Generated Ideas

To quantitatively understand the types of ideas that models generate during the execution-guided search, we perform a stratified analysis by classifying all generated ideas into either hyper-parameter tuning (including any ideas that can be implemented via changing existing configs) or algorithmic (including all ideas that involve implementing new changes not originally supported by the baseline codebase) by using an LLM-judge. Based on Table 2, all three models generate a substantial amount of algorithmic ideas apart from hyper-parameter tuning. Interestingly, different models exhibit different patterns, where Claude-4.5-Sonnet generates significantly more hyper-parameter ideas than both Claude-4.5-Opus and GPT-5. Moreover, the most effective ideas come from algorithmic ideas in most cases, except when using Claude-4.5-Sonnet. To complement the quantitative analysis, we provide several executed ideas in Table 3 and Appendix A.2. When sampling ideas, models would generate a thinking trace, followed by the natural language idea and a brief description of all the code changes needed to implement the idea. For brevity, we only include the natural language ideas in the table, but we present additional examples in Appendix A.3 with more details, including full code execution trajectories.

## 5. Reinforcement Learning from Execution Reward

Different from evolutionary search, reinforcement learning is an alternative learning algorithm that shapes model behaviors through gradient updates. Despite much recent success on verifiable domains like math and coding (DeepSeek-AI et al., 2025), RL's effectiveness on open-ended AI research

---

[1] https://github.com/KellerJordan/modded-nanogpt

*Table 2.* Breakdown of hyper-parameter tuning vs algorithmic ideas throughout the entire execution-guided search. We report the percentage of each type among all generated ideas of each model ($N = 500$ ideas on GRPO and $N = 800$ ideas on nanoGPT). We also report the average and best performance for ideas under each category, where we use validation accuracy as the performance metric for GRPO and validation loss as the metric for nanoGPT. Bold numbers every row indicate the best performance by each model. All models generate a substantial amount of algorithmic ideas apart from hyper-parameter changes, while Claude-4.5-Sonnet generates significantly more hyper-parameter ideas than other models.

| Model name | Hyper-parameter | | | Algorithmic | | |
|---|---|---|---|---|---|---|
| | Percentage | Average | Best | Percentage | Average | Best |
| | *GRPO environment (accuracy↑)* | | | | | |
| GPT-5 | 5.0% | 45.0% | 50.2% | 95.0% | 44.5% | **60.0%** |
| Claude-4.5-Sonnet | 41.1% | 48.4% | **69.4%** | 58.9% | 45.0% | 67.4% |
| Claude-4.5-Opus | 3.7% | 44.4% | 50.4% | 96.3% | 46.5% | **61.6%** |
| | *nanoGPT environment (loss↓)* | | | | | |
| GPT-5 | 15.4% | 3.254 | 3.195 | 84.6% | 3.894 | **3.170** |
| Claude-4.5-Sonnet | 31.3% | 3.251 | **3.208** | 68.7% | 3.679 | **3.208** |
| Claude-4.5-Opus | 8.7% | 3.329 | 3.147 | 91.3% | 3.419 | **3.141** |

remains unclear. For the first time, we explore whether we can leverage the automated executor as a reward function to directly finetune LLMs to generate more effective ideas via RL. We detail our implementation, experiment setup, and analysis of the training dynamics.

### 5.1. Reward Design and Experiment Setup

We use Qwen3-30B-A3B (Yang et al., 2025a) as the base model and finetune it using the standard GRPO algorithm (Shao et al., 2024), motivated by its consistent empirical success on other verified domains. Our prompt batch size is one since we only have one prompt for each environment. In the prompt, we provide the baseline codebase and ask the model to generate new ideas to improve the baseline.

We use large group sizes to stabilize training. For the post-training environment, we use a group size of 256; for the pre-training environment, we use a group size of 128. Since each idea on the GRPO environment runs on one single GPU and each idea on the nanoGPT environment runs on 8 GPUs, these group sizes correspond to parallel execution on 256 GPUs (for GRPO) or 1024 GPUs (for nanoGPT) to obtain the execution reward on each batch of rollout ideas. Each rollout is a thinking trace followed by the natural language idea. We set a max output length of 8192 tokens for rollout sampling and only feed the extracted ideas to the automated executor without the preceding thinking trace.

For the post-training environment, we use the validation set accuracy of each rollout idea after execution as the reward. For ideas without a valid accuracy (i.e., when the execution failed due to code generation errors), we assign a reward of 0. For the pre-training environment, we use the reciprocal of the validation loss as the reward ($\frac{1}{loss}$) and assign a reward of 0 to ideas with failed execution. Our experiments are based on Tinker (Thinking Machines Lab, 2025).

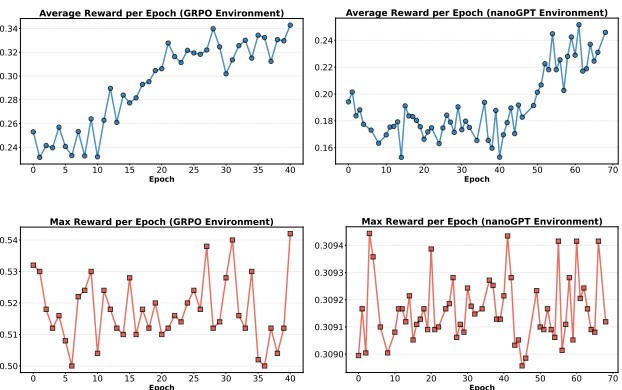

*Figure 5.* Training curves of RL from execution reward. We plot the average reward per epoch in the upper row, and the max reward per epoch in the lower row. For the GRPO environment, the reward is the accuracy; for the nanoGPT environment, the reward is the reciprocal of the loss. The average reward increases, but not the max reward.

### 5.2. Experiment Results

**Positive Training Curves for Average Reward** We plot the average reward of all rollouts of each training epoch in the upper row of Figure 5. For the first time, we demonstrate that the average performance of the generated ideas can increase after sufficient training epochs for open-ended research environments. For instance, the average accuracy on the GRPO environment increases from 0.253 at the beginning to 0.343 after 40 training epochs (top left plot of Figure 5); and the average reward on the nanoGPT environment increases from 0.194 at the beginning to 0.246 after 68 epochs (top right plot of Figure 5), corresponding to a decrease in the average validation loss from 5.150 to 4.066. Such curves echo prior one-shot RVLR gains in other verifiable domains like math (Wang et al., 2025).

| Claude-4.5-Opus on GRPO | Claude-4.5-Sonnet on GRPO |
|---|---|
| Residual Ratio Learning with Momentum Bounds: Instead of directly using the (importance sampling) ratio, decompose it into a "base" component (EMA of batch mean ratios) and a "residual" component (ratio – base). Apply sigmoid bounding only to the residual, allowing the base to capture systematic policy drift while controlling deviations from it. Combined with momentum clip adaptation. Formula: `residual = ratio – ema_batch_ratio,bounded_residual = sigmoid_bound(residual, deviation), effective_ratio = 1.0 + bounded_residual.`

**Validation Accuracy: 61.6**

Advantage Rank Difference Weighting: Instead of using absolute advantage magnitude, weight samples by how far their rank differs from their expected rank under uniform distribution. Samples that significantly outperform or underperform their "expected" position get higher weights. This is distribution-free and robust to outliers. Formula: `expected_rank = (N-1)/2, rank_diff = |actual_rank – expected_rank| / expected_rank, weight = 0.5 + 0.5 * rank_diff.`

**Validation Accuracy: 59.2** | Dynamic Mathematical Problem Difficulty Balancing with Performance Feedback: Implement intelligent difficulty balancing that dynamically adjusts the mix of problem difficulties based on recent performance trends. When performance is strong, increase difficulty proportion; when struggling, provide more foundational problems. Combine with the proven hyper-parameters for optimal learning progression.

**Validation Accuracy: 64.0**

Implement token-level reward attribution by using attention weights to identify which input tokens contributed most to correct answers, then amplifying the gradient updates for those tokens during policy gradient training.

**Validation Accuracy: 45.2**

Create mathematical working memory simulation by maintaining a context buffer of mathematical facts, definitions, and intermediate results during problem solving. This buffer gets updated as the model works through problems and provides additional context for subsequent mathematical steps, simulating how humans maintain mathematical working memory during complex calculations.

**Validation Accuracy: 58.0** |

*Table 3.* Examples of successfully executed ideas on the GRPO environment, along with their accuracy on the MATH validation set. The baseline accuracy is 48.0% on this environment.

**The Case of Max Reward**    Despite successfully reproducing the positive training curves observed in other domains, we argue that there is a distinction between idea generation and other verifiable domains. For advancing scientific discovery, we often care about the upper-bound of idea generation, rather than the average quality. In our particular case, we care more about having one breakthrough idea that dominates the baselines, rather than having many safe ideas with a high average. Thus, we plot the max reward of all rollouts at each training epoch in the lower row of Figure 5. The trend here is very different – **the max reward is fluctuating throughout RL training without a clear upward trend**. This reveals the crucial limitation of the standard GRPO algorithm for improving idea generation. In the next subsection, we analyze why RL from execution reward improves the average reward but not the max.

### 5.3. Analysis of Training Dynamics

**Thinking Length**    We first plot how the lengths of the thinking traces evolve over RL training in the upper row of Figure 6. In both environments, the thinking traces rapidly decrease in length while the idea lengths stay roughly constant. This is the opposite of the thinking emergence trend from prior RLVR work, such as DeepSeek-R1 (DeepSeek-AI et al., 2025). To further understand why the thinking length decreases, we investigate the correlation between the idea execution rate and the thinking trace length. In the bottom row of Figure 6, for the first 20 epochs of the RL

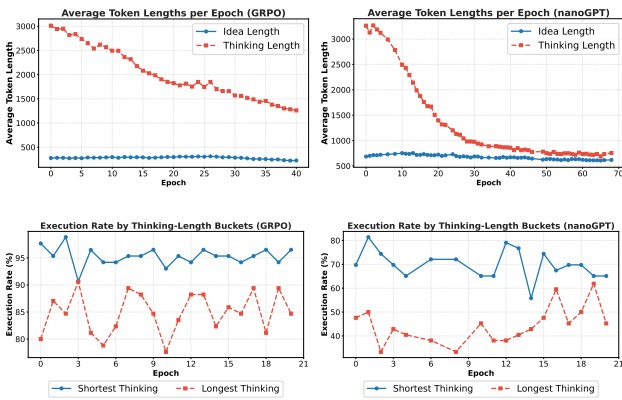

*Figure 6.* Upper Row: Average length of the thinking trace (red line) and the idea (blue line) per training epoch. Lower Row: Execution rate of ideas with the longest (red line) versus shortest (blue line) thinking traces. Ideas with longer thinking have lower execution rates. Correspondingly, thinking lengths decrease in RL.

training, we sort all ideas in each epoch by their thinking trace lengths and plot the average execution rate of the top-30% longest thinking ideas (red line) and the bottom-30% shortest thinking ideas (blue line). We see a clear trend where ideas with longer thinking consistently have a lower execution rate. We thus hypothesize that longer thinking correlates with more complex ideas with lower execution rates, leading the model to prefer shorter thinking instead in order to maximize the reward.

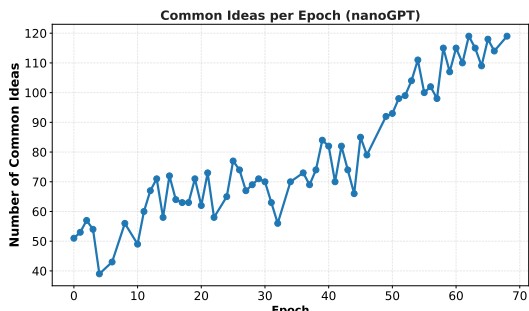

*Figure 7.* We plot how many ideas in each epoch are about either replacing RMSNorm with LayerNorm or doing EMA. The model converged on these two ideas after RL training.

**Diversity Collapse**  Upon manual investigation of all the rollouts being sampled throughout the RL training, we also observed a diversity collapse. Specifically, the model learned to converge on a few simple ideas that can consistently get a positive reward. For example, in the nanoGPT environment, the model learned to converge towards two common ideas: (1) replacing RMSNorm with LayerNorm; and (2) performing exponential moving average (EMA) over intermediate model checkpoints. As shown in Figure 7, out of a batch of 128 sampled ideas per epoch, 51 ideas sampled from Qwen3-30B at epoch 0 are one of the two common ideas above. Towards the end of the RL training, 119 out of 128 sampled ideas at epoch 68 are one of the two common ideas, indicating a severe diversity collapse.

The above analyses reveal that RL causes the models to converge on a few simple-to-implement ideas, accompanied by shrinking thinking lengths. These lead to an increase in the average reward, but do not push the upper-bound due to the lack of exploration. This phenomenon is analogous to mode-collapse observations on other verifiable domains, where the pass@k performance stagnates or even decreases after RL (Yue et al., 2025; Wu et al., 2025). Avoiding such convergence and collapse is an open problem and likely requires new algorithmic interventions beyond standard GRPO, which is beyond the scope of this work. However, we do share several preliminary attempts, including: sampling and appending previous epochs' trajectories into the current epoch's prompt for rollout sampling, adding a weighted length reward in the total reward, and adding a weighted similarity penalty in the total reward. We did not observe clear gains in the initial epochs and thus early-stopped them, but we document all these results in Appendix A.1 to inform future work.

## 6. Related Work

**AutoML**  Our work has deep connections to the AutoML literature. For example, the Neural Architecture Search (NAS) line of work typically defines a constrained set of neural network operators and optimizes for architectures based on validation set performance through reinforcement learning (Zoph & Le, 2017; Zoph et al., 2017) or search (Liu et al., 2018; So et al., 2019). More recent works explored directly using LLMs to propose architecture variants and implement them for validation (Liu et al., 2025; Cheng et al., 2025). Beyond architectures, similar automatic optimizations have been applied to improve hyperparameter tuning (Zhang et al., 2023), discover machine learning algorithms (Real et al., 2020), improve post-training objectives (Lu et al., 2024a), discover better neural network optimizers (Chen et al., 2023), and design agent scaffolds (Hu et al., 2025). Different from this line of work, we tackle automated AI research in a fully open-ended setting without any constraint on the type of ideas. Moreover, our goal is to improve the effectiveness of idea generation, where natural language ideas represent a higher level of abstraction than specific architecture variants or code optimizations.

**LLM-based Research Agents**  Recent works have been building end-to-end research agents (Lu et al., 2024b; Yamada et al., 2025; Tang et al., 2025; Schmidgall et al., 2025) that use LLMs to generate ideas and implement them through carefully designed scaffolds. They address open-ended AI research as we do, but do not study how to learn from execution feedback. On the other hand, on more grounded benchmarks with clear performance metrics such as MLE-Bench (Chan et al., 2025), RE-Bench (Wijk et al., 2024), and ML-Gym (Nathani et al., 2025), various works have explored how to learn from execution feedback through search (Toledo et al., 2025; Jiang et al., 2025) or RL (Yang et al., 2025b) to optimize performance on these targeted ML engineering tasks. While we also study algorithms for learning from execution feedback, we tackle open-ended research problems like pre-training and post-training rather than ML engineering tasks that heavily depend on feature engineering and hyper-parameter tuning.

**AI for Research**  Apart from fully end-to-end automated AI research, many works have studied how to use LLMs for specific components of the scientific research pipeline, such as literature review (Asai et al., 2024; L'ala et al., 2023), idea generation (Si et al., 2025b; Wang et al., 2024), data analysis (Majumder et al., 2025; Mitchener et al., 2025), experiment plan generation (Goel et al., 2025), research code execution (Starace et al., 2025; Hua et al., 2025; Tian et al., 2024), and paper reviewing (Liang et al., 2024; Zhu et al., 2025). Our work focuses on automated idea execution and learning from the execution feedback. We consider our work complementary to many of the above works that improve other aspects of the scientific research pipeline.

**Execution Grounding for Code**  The idea of learning from execution feedback has been explored in the code

generation domain. For example, Zheng et al. (2024) curate data and train models to refine code from either human or execution feedback; Gehring et al. (2025) use end-to-end RL training to teach models to improve code based on execution feedback; Lavon et al. (2025) directly guide code generation with execution signals during inference time. In contrast, our work explores execution grounding for the application of idea generation, where the verification is more complicated and expensive.

## 7. Conclusion

In this work, we built a large-scale parallel executor for automatically executing model-generated ideas to verify their effectiveness on open-ended LLM research problems, including both pre-training and post-training. Using this executor as a reward function, we analyzed the effectiveness of execution-guided evolutionary search, where frontier LLMs equipped with our search scaffold can significantly outperform the baseline solutions. We also analyzed the limitations of reinforcement learning with execution rewards, where models converge on simple ideas to improve the average reward but lose diversity and do not improve the upper-bound. Our empirical results demonstrate the feasibility and potential of the automated execution feedback loop and also point out the remaining limitations for future improvement.

## Discussion

Despite encouraging initial signals, there are still many limitations to our current set of experiments, which would be great directions for future improvement.

First, our current procedure does not test the generalizability of the generated ideas. It is possible that the best-performing ideas at the small scales may not transfer to gains at a larger scale or on other datasets. Future work should explore methods that explicitly test such generalizability and scalability, and even incorporate them as part of the optimization objectives.

Second, we have shown that RL with execution reward in our current setup can only improve the average reward but not the upper bound. There are many possible reasons, such as a lack of diversity in the base model and missing exploration incentives in the current RL objective. Future work should explore remedies and better learning algorithms for LLMs to more efficiently learn from the execution feedback. For instance, future works could explore how to exploit richer learning signals from the execution trajectories beyond just scalar rewards.

Third, our current experiment scope is bounded by the capability of the execution agent. There exist many promising model-generated ideas that could not be successfully executed by the execution agent (e.g., see the end of Appendix A.2), leading to noise in the reward signal. Future work could develop more capable execution agents and extend our setup to even more complex research problems. For instance, instead of directly prompting an LLM for code diff, future work can implement more capable coding agents with access to external tools and the ability to install new libraries in the execution environments.

Last but not least, we only explored effectiveness as the training reward in this work. There are many other, more subjective alternative metrics that could complement the effectiveness reward, such as the idea novelty and interestingness. Future work could explore how to computationally measure them and incorporate them as part of the training objective to discover more insightful ideas.

## Impact Statement

We intend this work to advance the field of AI research, which in turn may benefit scientific progress across a broad range of domains, including the natural sciences, medicine, and engineering. At the same time, we recognize the importance of proper governance and oversight to ensure that such systems are developed and deployed responsibly. We encourage the community to invest in frameworks for auditing and interpreting the outputs of automated research pipelines, and to explore effective ways for human oversight and human-AI collaboration.

## Acknowledgment

We thank DST Global, Laude Institute, and Thinking Machines Lab for their generous sponsorship of computing resources. We thank Yuandong Tian, Edward Hughes, Ludwig Schmidt, Cong Lu, Jenny Zhang, Jiaxin Wen, Chris Rytting, Chen Zhao, Xinran Zhao, Yanzhe Zhang, John Yang, Shicheng Liu, Andy Zhou, Will Held, Haotian Ye, Luke Bailey, as well as members of Tatsu Lab and SALT Lab for their helpful discussion and feedback. This work was supported by an HAI grant, DSO labs, gifts from Open Philanthropy, Amazon, Schmidt Sciences, the Tianqiao and Chrissy Chen Foundation and a grant under the NSF CAREER IIS-2338866 and IIS-2247357, ONR N00014-24-1-2609 and N00014-24-1-2532, and DARPA Cooperative Agreement HR00112520013. This work does not necessarily reflect the position or policy of the government and no official endorsement should be inferred.

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

# A. Appendix

## A.1. Other RL Attempts

We present several attempts to improve our RL from the execution reward recipe.

**Attempt 1: Dynamic Prompt**    At each epoch (except the first epoch), we randomly sample different executed idea trajectories from the previous epoch and append them to the idea sampling prompt when sampling new rollouts. This merges in-context learning with RL and adds diversity to the idea sampling process. We present the experiment results on the GRPO environment in Figure 8. We did not see significant improvement in early epochs and thus early stopped.

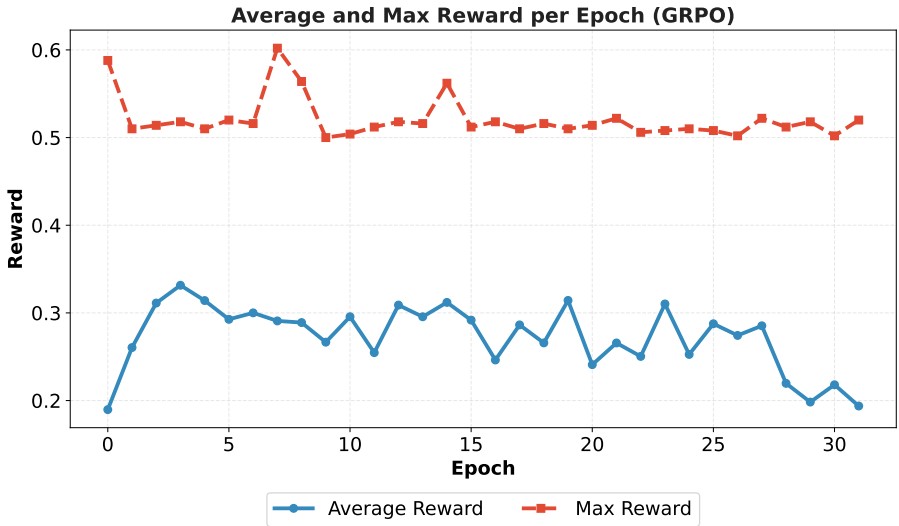

*Figure 8.* RL with dynamic prompt by appending trajectories from previous epochs. Results are on the GRPO environment.

**Attempt 2: Length Reward**    Since we noted a rapid thinking length decrease in our main RL experiment, we tried a simple fix by adding a weighted length reward that counts the number of tokens in the entire rollout sequence, including the thinking trace and the idea. We cap the length reward to a maximum of $0.3$ to avoid it dominating the accuracy reward. We present the experiment results on the GRPO environment in Figure 9. As shown on the right panel, the thinking length no longer decreases after adding the length reward to the total reward, but the total training reward isn't going up as shown on the left panel.

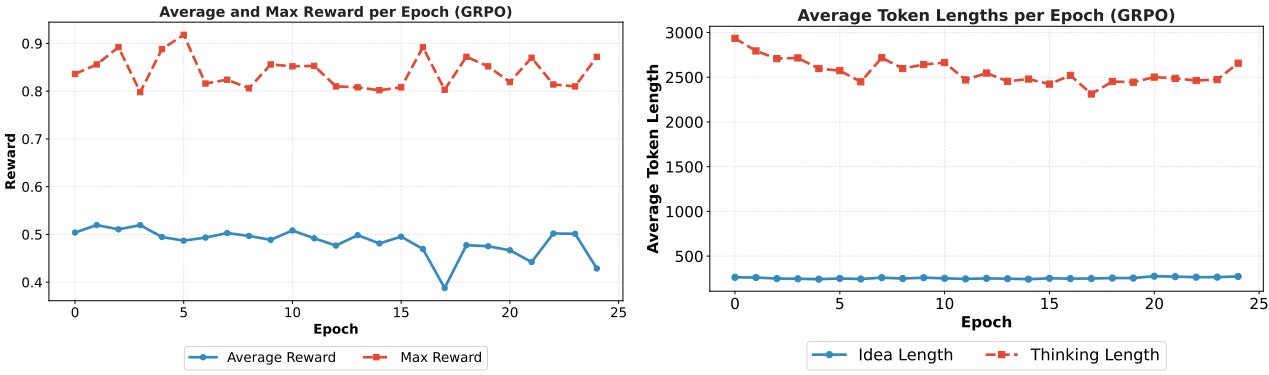

*Figure 9.* Average and max reward throughout RL training when adding the length reward (left), as well as the progression of the thinking and idea lengths (right).

**Attempt 3: Diversity Reward**    We also tried adding a diversity reward in addition to the effectiveness reward. Specifically, when computing the reward for each rollout, we compute its token-level Jaccard similarity with ideas from the previous epoch and add the negative similarity as a penalty to the total reward to discourage repeating ideas that have already been generated before. In fact, this is similar to one of the post-training ideas proposed by Claude-4.5-Sonnet (see example 4 in Appendix A.3). We show the training curves on the GRPO environment in Figure 10. The model maintains a consistent idea similarity (right plot), suggesting sustained exploration. The effectiveness reward is generally showing an upward trend (left plot), but not markedly better than the main RL run with just the effectiveness reward (first sub-plot in Figure 5).

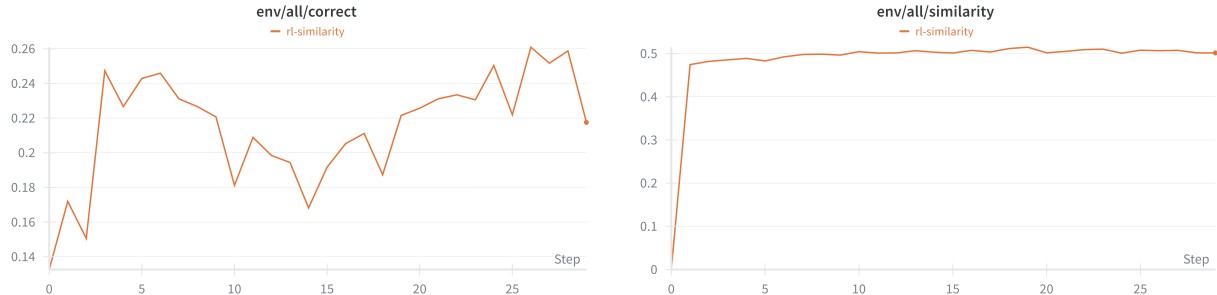

*Figure 10.* Effectiveness reward (left) and average idea similarity to previous epoch (right) when doing RL with diversity reward.

## A.2. Additional Idea Examples

We provide several additional example ideas generated by Claude-4.5-Opus (Table 4) and Claude-4.5-Sonnet (Table 5) on the GRPO environment, including ideas with failed code execution. In most cases, code execution errors happen when the idea involves complicated changes or installing and importing external packages not supported in our execution environment. Future work should explore improvement to the execution agent so that more complicated types of ideas (e.g., training additional auxiliary models or system-level optimizations) can be implemented correctly.

| Successful Execution | Failed Execution |
|---|---|
| **[Experiment]** Sequence Position Weighted Trust Region: Apply tighter sigmoid bounds to earlier tokens in the sequence (where errors compound) and looser bounds to later tokens. Weight: `position_weight = 1 - 0.3 * (position / seq_len)`, `effective_deviation = 0.25 + 0.2 * position_weight`. This accounts for the sequential nature of autoregressive generation. 

 **[Code Changes]** Modify `grpo.py`: Initialize `current_cliprange = 0.2`, `ema_clip_fraction = 0.15`. Standard momentum clip updates. Modify `compute_grpo_clip_loss` in `grpo_utils.py`: After computing ratio on line 91 (shape: `batch_size x seq_len`): `batch_size, seq_len = ratio.shape`, `positions = torch.arange(seq_len, device=ratio.device).float().unsqueeze(0).expand(batch_size, -1)`, `position_weight = 1.0 - 0.3 * (positions / (seq_len - 1 + 1e-6))`, `effective_deviation = 0.25 + 0.2 * position_weight`. Apply position-aware sigmoid: `centered_ratio = ratio - 1.0`, `bounded_ratio = 1.0 + (2.0 * torch.sigmoid(centered_ratio) - 1.0) * effective_deviation`. Use: `surr1 = bounded_ratio * advantages`, `surr2 = torch.clamp(bounded_ratio, 1 - cliprange, 1 + cliprange) * advantages`, `loss = -torch.min(surr1, surr2)`. Add metadata: `metadata["mean_effective_deviation"] = effective_deviation.mean().item()`, `metadata["early_deviation"] = effective_deviation[:, :seq_len//4].mean().item()`, `metadata["late_deviation"] = effective_deviation[:, -seq_len//4:].mean().item()`. 

 **Validation Accuracy: 59.8** | **[Experiment]** Hierarchical Position-Group Trust Region: Apply trust region at two hierarchical levels – group level (shared within each response group) and position level (varies along sequence). Groups with high internal reward variance get tighter group-level bounds. Within groups, positions follow the proven decay pattern. This captures both cross-sample and within-sample structure. Formula: `group_dev = 0.4 - 0.15 * tanh(group_reward_var / 0.3)`, `position_factor = 1 - 0.2 * rel_pos`, `effective_dev = group_dev * position_factor`. 

 **[Code Changes]** Modify `grpo.py`: Initialize `current_cliprange = 0.2`, `ema_clip_fraction = 0.15`. Standard momentum clip updates. Pass `group_size` to function. Modify `compute_grpo_clip_loss` in `grpo_utils.py`: Add parameter `group_size=8`. After computing ratio: `batch_size, seq_len = ratio.shape`, `n_groups = batch_size // group_size`. Compute group reward variance from advantages as proxy: `adv_grouped = advantages.view(n_groups, group_size, -1)`, `group_adv_var = adv_grouped.var(dim=1, keepdim=True)`, `group_adv_var_expanded = group_adv_var.expand(-1, group_size, -1).reshape(advantages.shape)`. Group-level deviation: `group_deviation = 0.4 - 0.15 * torch.tanh(group_adv_var_expanded / 0.3)`. Position factor: `positions = torch.arange(seq_len, device=ratio.device).float().unsqueeze(0).expand(batch_size, -1)`, `rel_pos = positions / (seq_len - 1 + 1e-6)`, `position_factor = 1.0 - 0.2 * rel_pos`. Hierarchical deviation: `effective_deviation = group_deviation * position_factor`, `effective_deviation = torch.clamp(effective_deviation, 0.15, 0.45)`. Apply: `centered_ratio = ratio - 1.0`, `bounded_ratio = 1.0 + (2.0 * torch.sigmoid(centered_ratio) - 1.0) * effective_deviation`. Use: `surr1 = bounded_ratio * advantages`, `surr2 = torch.clamp(bounded_ratio, 1 - cliprange, 1 + cliprange) * advantages`, `loss = -torch.min(surr1, surr2)`. Add `metadata["mean_group_var"] = group_adv_var.mean().item()`, `metadata["mean_effective_deviation"] = effective_deviation.mean().item()`. Log to wandb. |

*Table 4.* Additional examples on the GRPO environment. Ideas are generated by Claude-4.5-Opus during evolutionary search.

| Successful Execution | Failed Execution |
|---|---|
| **[Experiment]** Create a mathematical step-complexity aware reward shaping where responses with more mathematical reasoning steps receive slightly higher base rewards (1.05x for 3+ steps, 1.1x for 5+ steps) when correct, encouraging thorough mathematical exposition without changing the core binary reward structure. | **[Experiment]** Implement temporal difference advantage estimation where advantages incorporate not just current rewards but also predicted future rewards using a learned value function, combined with the proven `3e-5` learning rate and `reinforce_with_baseline` for more accurate credit assignment. |
| **[Code Changes]** Modify `r1_zero_reward_fn_train` in `drgrpo_grader.py` to count reasoning steps by detecting mathematical transitions ("therefore", "thus", "so", "=", "=>"). When answer is correct, apply step-based multiplier: `step_multiplier = 1.0 + 0.05 * min(2, max(0, num_steps - 2))` to get multipliers `[1.0, 1.05, 1.1]`. Return `{"format_reward": 1.0, "answer_reward": answer_reward, "reward": base_reward * step_multiplier}`. Set `-learning_rate 3e-5` and `-loss_type reinforce_with_baseline`. | **[Code Changes]** Add a value head to the policy model in `grpo.py` that predicts expected future rewards. Compute TD advantages as `advantages = rewards + gamma * next_values - current_values` with `gamma=0.99`. Train the value function with MSE loss on observed returns. Modify `compute_group_normalized_rewards` to use TD advantages instead of basic reward differences. Set `-learning_rate 3e-5` and `-loss_type reinforce_with_baseline`. |
| **Validation Accuracy: 65.6** | |
| **[Experiment]** Add experience replay by maintaining a buffer of the top 20% highest-reward rollouts from previous epochs and mixing them (25% replay, 75% new) with current rollouts during training, combined with `3e-5` learning rate and `reinforce_with_baseline` for improved sample efficiency. | **[Experiment]** Ensemble Decision Training with Voting Consensus: Train the model using ensemble-style decision making where each rollout generates multiple candidate responses, and the final training signal is based on majority voting among responses. This encourages the model to develop more robust and consistent reasoning patterns while maintaining diversity in solution approaches. |
| **[Code Changes]** Modify `train_loop` in `grpo.py` to maintain `replay_buffer` storing high-reward (`>0.8`) rollouts from previous epochs. Each epoch, sample 25% of training data from replay buffer and 75% from new rollouts. Update buffer by adding top 20% of current epoch's rollouts and removing oldest entries when buffer exceeds 1000 samples. Set `-learning_rate 3e-5` and `-loss_type reinforce_with_baseline`. | **[Code Changes]** Modify `sample_rollout` in `sample.py` to generate 3 responses per prompt instead of 1, using different random seeds. Implement voting consensus in `r1_zero_reward_fn_train`: if 2+ responses are correct, apply `+0.15` consensus bonus; if responses disagree, apply `-0.05` uncertainty penalty. In `train_loop` in `grpo.py`, select the highest-voted response for training while using consensus information to adjust learning rate: `consensus_lr = 3e-5 * (0.9 + 0.2 * consensus_rate)`. Set `group_size=6`, `-loss_type reinforce_with_baseline`. |
| **Validation Accuracy: 39.4** | |
| **[Experiment]** Implement response diversity rewards within groups where responses to the same prompt receive bonus rewards (`0.05-0.15`) for being dissimilar to other responses in their group, encouraging exploration of different solution paths while maintaining the proven `group_size=8` and `3e-5` learning rate combination. | **[Experiment]** Implement hierarchical advantage estimation where advantages are computed at both token-level and sequence-level, with token-level advantages weighted by their position importance (higher weights for mathematical expressions and final answers), combined with the successful `3e-5` learning rate and `reinforce_with_baseline`. |
| **[Code Changes]** Modify `compute_group_normalized_rewards` in `grpo_utils.py` to compute pairwise similarity between responses in each group using token-level Jaccard similarity. Add diversity bonus: `diversity_reward = 0.15 * (1 - max_similarity_in_group)` to each response's reward before advantage computation. Reshape responses into groups, compute similarities, and add bonuses before advantage normalization. Set `-learning_rate 3e-5`, `-loss_type reinforce_with_baseline`, `-group_size 8`. | **[Code Changes]** Modify `grpo_microbatch_train_step` in `grpo_utils.py` to create position importance weights that assign `2.0x` weight to tokens containing mathematical symbols (`\frac`, `+`, `-`, `*`, `=`) and `1.5x` weight to answer sections. Compute both sequence-level advantages (current) and token-level advantages, then combine as `final_advantages = 0.6 * sequence_advantages + 0.4 * token_advantages`. Set `-learning_rate 3e-5` and `-loss_type reinforce_with_baseline`. |
| **Validation Accuracy: 19.2** | |

*Table 5.* Additional examples on the GRPO environment. Ideas are generated by Claude-4.5-Sonnet during evolutionary search.

We also present the top-performing ideas from Claude-4.5-Opus, Claude-4.5-Sonnet, and GPT-5 on the nanoGPT environment below:

---

**Claude-4.5-Opus Idea on nanoGPT (Validation Loss: 3.1407)**

**[Experiment]** Wider SwiGLU (5x) with MLP Output Scaling (Init 0.97), Skip Connections Every 4 and 8 Layers with Learnable Weights (Init 0.52 and 0.31), Separate Attention/MLP Scales (Init 0.98), Higher LR (0.00168), Reduced Weight Decay (0.065), Warmup 173 iters, Lower Min LR (0.03x), Cosine Annealing, EMA, Untied Embeddings, and Beta2=0.99

Make the dual skip connection weights learnable parameters initialized at proven good values. This allows the model to adapt skip weights during training while combining with separate attention/MLP residual scales.

**[Code Changes]**

- Change `warmup_iters = 256` to `warmup_iters = 173` in Hyperparameters class

- Change `weight_decay = 0.1` to `weight_decay = 0.065` in Hyperparameters class

- Change `learning_rate = 0.0015` to `learning_rate = 0.00168` in Hyperparameters class

- In `GPT.__init__`, add after transformer dict:

  ```
  self.skip_weight_4 = nn.Parameter(torch.tensor(0.52))
  self.skip_weight_8 = nn.Parameter(torch.tensor(0.31))
  ```

- In `Block.__init__`, add: `self.attn_scale = nn.Parameter(torch.tensor(0.98))` and `self.mlp_scale = nn.Parameter(torch.tensor(0.98))`

- In `Block.forward`, change to:

  ```
  def forward(self, x):
      x = x + self.attn_scale * self.attn(rmsnorm(x))
      x = x + self.mlp_scale * self.mlp(rmsnorm(x))
      return x
  ```

- In `Block.forward_with_cache`, change to:

  ```
  def forward_with_cache(self, x, cache):
      attn_out, new_cache = self.attn.forward_with_cache(rmsnorm(x),
      cache=cache)
      x = x + self.attn_scale * attn_out
      x = x + self.mlp_scale * self.mlp(rmsnorm(x))
      return x, new_cache
  ```

- In `MLP.__init__`, replace lines 81–82 with:

  ```
  self.c_fc = nn.Linear(config.n_embd, 5 * config.n_embd, bias=False)
  self.c_gate = nn.Linear(config.n_embd, 5 * config.n_embd, bias=False)
  self.c_proj = nn.Linear(5 * config.n_embd, config.n_embd, bias=False)
  self.output_scale = nn.Parameter(torch.tensor(0.97))
  ```

- In `MLP.forward`, replace with:

  ```
  def forward(self, x):
      gate = F.silu(self.c_gate(x))
      x = self.c_fc(x) * gate
      x = self.c_proj(x) * self.output_scale
      return x
  ```

---

- In GPT.\_\_init\_\_, remove line 132: `self.transformer.wte.weight = self.lm_head.weight`

- Remove line 131: `self.lm_head.LLMC_SKIP_INIT = 1`

- Modify `_init_weights` to add: `if isinstance(module, nn.Linear): torch.nn.init.normal_(module.weight, mean=0.0, std=0.02)`

- Change optimizer betas on line 402 to `betas=(0.9, 0.99)`

- Modify `get_lr` function:

```
def get_lr(it):
    assert it <= args.num_iterations
    if it < args.warmup_iters:
        return args.learning_rate * (it+1) / args.warmup_iters
    min_lr = 0.03 * args.learning_rate
    decay_ratio = (it - args.warmup_iters) /
    (args.num_iterations - args.warmup_iters)
    return min_lr + 0.5 * (args.learning_rate - min_lr) *
    (1.0 + math.cos(math.pi * decay_ratio))
```

- In `GPT.forward`, replace the block loop with:

```
layer_outputs = []
for i, block in enumerate(self.transformer.h):
    if i >= 4 and i % 4 == 0:
        x = x + self.skip_weight_4 * layer_outputs[i-4]
    if i >= 8 and i % 8 == 0:
        x = x + self.skip_weight_8 * layer_outputs[i-8]
    x = block(x)
    layer_outputs.append(x)
```

- In `GPT.forward_with_cache`, replace the block loop with:

```
layer_outputs = []
for i, block in enumerate(self.transformer.h):
    if i >= 4 and i % 4 == 0:
        x = x + self.skip_weight_4 * layer_outputs[i-4]
    if i >= 8 and i % 8 == 0:
        x = x + self.skip_weight_8 * layer_outputs[i-8]
    x, new_cache = block.forward_with_cache(x, cache=caches[i])
    new_caches.append(new_cache)
    layer_outputs.append(x)
```

- After model initialization, add: `ema_model = {k: v.clone() for k, v in raw_model.state_dict().items()}` and `ema_decay = 0.999`

- After `optimizer.step()`, add: `for k, v in raw_model.state_dict().items(): ema_model[k].mul_(ema_decay).add_(v, alpha=1-ema_decay)`

- Before validation, add: `orig_state = {k: v.clone() for k, v in raw_model.state_dict().items()}; raw_model.load_state_dict(ema_model)`

- After validation, add: `raw_model.load_state_dict(orig_state)`

**Claude-4.5-Sonnet Idea on nanoGPT (Validation Loss: 3.2081)**

[Experiment] Two-phase weight decay (0.1170→0.0210 at 59.65%) + 30.45% plateau + LR 0.001550 + warmup 197 + two-phase grad clip (1.054→0.916 at 59.65%) + quadratic min_lr interpolation (0.0113x at 59.65%, 0.0075x at end via quadratic) + progressive EMA (0.999→0.9992 linear over training) + exponential warmup + cosine LR + beta2=0.99

Use smooth quadratic interpolation for `min_lr` during low-WD phase AND progressive EMA that gradually increases from `0.999` to `0.9992` linearly throughout training. Early training benefits from faster EMA tracking, while later training gets heavier smoothing. Use conservative settings: WD `0.1170/0.0210`, extended plateau `30.45%`, moderate LR `0.001550`, longest warmup `197`, tight grad clip `1.054→0.916`.

[Code Changes] Modify line 326 to change `warmup_iters = 256` to `warmup_iters = 197`. Modify line 325 to change `learning_rate = 0.0015` to `learning_rate = 0.001550`. Modify line 402 to change `betas=(0.9, 0.95)` to `betas=(0.9, 0.99)`. Modify the `get_lr` function: replace lines 408–414 with:

```
if it < args.warmup_iters:
    progress = (it + 1) / args.warmup_iters
    return args.learning_rate * (1.0 - math.exp(-5.0 * progress))
plateau_end = int(0.3045 * args.num_iterations)
if it < plateau_end:
    return args.learning_rate
overall_progress = it / args.num_iterations
decay_ratio = (it - plateau_end) / (args.num_iterations - plateau_end)
coeff = 0.5 * (1.0 + math.cos(math.pi * decay_ratio))
if overall_progress <= 0.5965:
    min_lr_factor = 0.0113
else:
    phase2_progress = (overall_progress - 0.5965) / (1.0 - 0.5965)
    min_lr_factor = 0.0113 - (0.0113 - 0.0075) * (phase2_progress ** 2)
min_lr = min_lr_factor * args.learning_rate
return min_lr + coeff * (args.learning_rate - min_lr)
```

Modify line 527 to:

```
progress = step / args.num_iterations
current_clip = 0.916 if progress > 0.5965 else 1.054
norm = torch.nn.utils.clip_grad_norm_(model.parameters(), current_clip)
```

After line 529, add:

```
progress = step / args.num_iterations
current_wd = 0.0210 if progress > 0.5965 else 0.1170
for param_group in optimizer.param_groups:
    param_group['weight_decay'] = current_wd
```

After line 387, add:

```
ema_model = {name: param.clone().detach() for name, param in
raw_model.named_parameters()}
```

After line 533, add:

```
if step > 0:
    progress = step / args.num_iterations
    ema_decay = 0.999 + 0.0002 * progress
    for name, param in raw_model.named_parameters():
        ema_model[name].mul_(ema_decay).add_(param.data, alpha=1 - ema_decay)
```

Before line 483, add:

```
original_params = {name: param.data.clone() for name, param in
```

```
raw_model.named_parameters()}
for name, param in raw_model.named_parameters():
    param.data.copy_(ema_model[name])
```

After line 509, add:

```
for name, param in raw_model.named_parameters():
    param.data.copy_(original_params[name])
```

---

### GPT-5 Idea on nanoGPT (Validation Loss: 3.1697)

**[Experiment]** SwiGLU-3.5x + Residual Alphas + Min-Floor Cosine + Per-step Beta2 Linear Decay + 3-Group AdamW + Debiased EMA

**[Code Changes]**

- **Hyperparameters:** `hidden_factor=3.5`, `warmup_iters=256`, `lr_peak_factor=1.10`, `min_lr_factor=0.02`, `beta2_start=0.99`, `beta2_end=0.95`, `wd_decay=0.1`, `wd_embed=0.01`, `ema_decay=0.9995`, `ema_warmup_steps=256`.

- **MLP:** SwiGLU; Block alphas init `0.9`.

- **Optimizer:** 3-group AdamW.

- **LR:** warmup to peak; cosine to floor as before.

- **Per-step beta2 update:** After setting lr each step, set

$$\texttt{beta2} = \texttt{beta2\_start} + \left(\texttt{beta2\_end} - \texttt{beta2\_start}\right) \min\left(1.0, \frac{\texttt{it} + 1}{\texttt{args.num\_iterations}}\right);$$

  update all `param_groups` betas.

- **EMA:** maintain `ema_params` with debiasing at eval (divide by `1 - ema_decay**ema_step`), then restore.

While the best-performing ideas on nanoGPT tend to be heavily optimized with extensive hyper-parameter tuning mixed with various architecture tweaks, we also pick a few more "atomic" algorithmic ideas that are successfully executed.

**Examples from Claude-4.5-Opus**

- **Head-Wise Attention Output Scaling** Add learnable per-head scaling factors to attention, allowing different heads to contribute with different magnitudes to the output.
  **Validation Loss: 3.2386**

- **Learned Residual Connection Weights** Add learnable scalar weights for each residual connection that are initialized to 1.0, allowing the model to learn optimal residual scaling during training.
  **Validation Loss: 3.2517**

- **Mixture of Embeddings with Position** Learn to mix token embeddings and position embeddings with a content-dependent weight, allowing the model to dynamically balance positional vs semantic information per token.
  **Validation Loss: 3.2497**

- **Shared Input-Output Embedding with Learned Asymmetry** Keep weight tying but add a small learned transformation on the output side, providing the benefits of weight tying while allowing output-specific adaptation.
  **Validation Loss: 3.2499**

- **Gated Final Normalization** Replace the final RMSNorm before `lm_head` with a gated version where a learned gate controls how much normalization is applied vs passing the raw representation.
  **Validation Loss: 3.2503**

- **Position-Aware MLP Gating** Gate the MLP output based on position information, allowing the model to learn position-dependent processing depth.
  **Validation Loss: 3.2506**

- **Learned Residual Connection Weights** Add learnable scalar weights for each residual connection that are initialized to 1.0, allowing the model to learn optimal residual scaling during training.
  **Validation Loss: 3.2517**

- **Grouped Token Embeddings** Group the vocabulary into clusters and add a learned embedding per cluster on top of token embeddings, providing hierarchical vocabulary structure.
  **Validation Loss: 3.2521**

Lastly, we present several interesting ideas on the nanoGPT environment that didn't get successfully executed. These examples are generated by Claude-4.5-Opus.

- **Soft Layer Repetition** Allow the model to softly repeat computation through layers by adding a learned gate that mixes the current layer's input back into its output, simulating variable depth.

- **Causal Context Compression** Before each attention layer, apply a learned compression that mixes local context (previous 2-3 tokens) into the current representation, providing implicit local context without convolutions.

- **Attention Head Specialization via Orthogonal Loss** Add a soft penalty that encourages different attention heads to attend to different patterns by penalizing similarity between head outputs.

- **Skip Connections with Learned Residual Weights** Combine skip connections with learned residual weights. The skip connections provide alternative gradient paths while learned weights allow adaptive scaling.

- **Token Difficulty-Aware Loss Weighting** Weight the loss contribution of each token based on the model's uncertainty (entropy) at that position, focusing learning on difficult tokens while not over-optimizing easy ones.

## A.3. Code Execution Examples

We present a few ideas with their full code execution to demonstrate the full end-to-end trajectories. All examples below are from Claude-4.5-Sonnet on the GRPO environment. For each example, we first present the natural language idea, followed by the code implementation generated by Claude-4.5-Sonnet.

### Example 1

**[Experiment]** Create mathematical working memory simulation by maintaining a context buffer of mathematical facts, definitions, and intermediate results during problem solving. This buffer gets updated as the model works through problems and provides additional context for subsequent mathematical steps, simulating how humans maintain mathematical working memory during complex calculations.

**[Code Changes]** Modify the prompt template in `sample.py` to include a "Mathematical Context" section that accumulates relevant mathematical facts during problem solving. Implement context buffer management in `train_loop` in `grpo.py` that maintains recent mathematical definitions, formulas, and intermediate calculations across problems in the same batch. Update the buffer with key mathematical expressions extracted from each response. Use enhanced prompts that include relevant context from the buffer. Set `–learning_rate 2.5e-5` and `–loss_type reinforce_with_baseline`.

**Validation Accuracy: 58.0**

```
1  --- repo_variants_full_pipeline_bsz50_fixed2_epoch7/idea_43/sample.py
2  +++ repo_variants_full_pipeline_bsz50_fixed2_epoch7/idea_43/sample.py
3  @@ -2,6 +2,7 @@ from vllm import LLM, SamplingParams
4   import json
5   from tqdm import tqdm
6   import random
7  +import re
8
9   def load_prompt_template(prompt_path = "prompts/r1_zero.prompt"):
10      with open(prompt_path, 'r') as f:
11  @@ -20,12 +21,52 @@ def get_model_and_sampling_params(model_path = "Qwen/Qwen2.5-
       Math-1.5B", token
12
13       return llm, sampling_params
14
15  -def load_dataset_with_prompt_template(prompt_template, dataset_path="/juice5b/
       scr5b/nlp/aihinton/CS336_data/a5-alignment/MATH/test.jsonl"):
16  +def extract_mathematical_expressions(text):
17  +    """Extract mathematical expressions, formulas, and key facts from text."""
18  +    expressions = []
19  +
20  +    # Extract LaTeX math expressions
21  +    latex_patterns = [
22  +        r'\$[^$]+\$',   # Inline math
23  +        r'\\[a-zA-Z]+\{[^}]+\}',   # LaTeX commands
24  +        r'\\frac\{[^}]+\}\{[^}]+\}',   # Fractions
25  +        r'\\sqrt\{[^}]+\}',   # Square roots
26  +        r'[a-zA-Z]\s*=\s*[^,\n\.]+',   # Variable assignments
27  +    ]
28  +
29  +    for pattern in latex_patterns:
30  +        matches = re.findall(pattern, text)
31  +        expressions.extend(matches)
32  +
33  +    # Extract definitions and key facts
34  +    definition_patterns = [
35  +        r'[A-Z][a-z]+ is defined as [^\.]+\.',
36  +        r'Let [a-zA-Z] be [^\.]+\.',
```

```
37  +            r'Given that [^\.]+\.',
38  +            r'Since [^\.]+\.',
39  +            r'Therefore [^\.]+\.',
40  +        ]
41  +
42  +    for pattern in definition_patterns:
43  +        matches = re.findall(pattern, text)
44  +        expressions.extend(matches)
45  +
46  +    return list(set(expressions))  # Remove duplicates
47  +
48  +def create_context_prompt(base_template, context_items):
49  +    """Create a prompt with mathematical context."""
50  +    if context_items:
51  +        context_section = "Mathematical Context:\n" + "\n".join(f"- {item}" for
        item in context_items[-10:]) + "\n\n"
52  +        return context_section + base_template
53  +    return base_template
54  +
55  +def load_dataset_with_prompt_template(prompt_template, dataset_path="/juice5b/
        scr5b/nlp/aihinton/CS336_data/a5-alignment/MATH/test.jsonl", context_items=
        None):
56      with open(dataset_path, 'r') as f:
57          dataset = [json.loads(line) for line in f]
58      print(f"Loaded {len(dataset)} examples from {dataset_path}")
59
60  -    prompts = [prompt_template.format(question=example["problem"]) for example in
        dataset]
61  +    enhanced_template = create_context_prompt(prompt_template, context_items)
62  +    prompts = [enhanced_template.format(question=example["problem"]) for example
        in dataset]
63      answers = [example["answer"] for example in dataset]
64      return prompts, answers
65
66  @@ -93,4 +134,42 @@
67      if return_rewards:
68          return batched_prompts, batched_answers, batched_responses,
        batched_rewards
69      return results
70  -
71  +
72  +class MathContextBuffer:
73  +    """Buffer to maintain mathematical working memory across problems."""
74  +
75  +    def __init__(self, max_size=50):
76  +        self.buffer = []
77  +        self.max_size = max_size
78  +
79  +    def add_expressions(self, expressions):
80  +        """Add new mathematical expressions to the buffer."""
81  +        for expr in expressions:
82  +            if expr and len(expr.strip()) > 3:  # Filter out very short
        expressions
83  +                self.buffer.append(expr.strip())
84  +
85  +        # Keep only the most recent items
86  +        if len(self.buffer) > self.max_size:
87  +            self.buffer = self.buffer[-self.max_size:]
88  +
89  +    def get_relevant_context(self, query_text, max_items=10):
90  +        """Get the most relevant context items for a given query."""
91  +        if not self.buffer:
92  +            return []
```

```
 93  +
 94  +            # Simple relevance scoring based on shared mathematical terms
 95  +            scored_items = []
 96  +            query_lower = query_text.lower()
 97  +
 98  +            for item in self.buffer[-20:]:  # Consider recent items
 99  +                score = sum(1 for word in item.lower().split() if word in query_lower
         )
100  +                if score > 0:
101  +                    scored_items.append((score, item))
102  +
103  +            # Return top items sorted by relevance
104  +            scored_items.sort(reverse=True, key=lambda x: x[0])
105  +            return [item[1] for item in scored_items[:max_items]]
106  +
107  +      def clear(self):
108  +            """Clear the context buffer."""
109  +            self.buffer = []
110  --- repo_variants_full_pipeline_bsz50_fixed2_epoch7/idea_43/grpo.py
111  +++ repo_variants_full_pipeline_bsz50_fixed2_epoch7/idea_43/grpo.py
112  @@ -1,6 +1,6 @@
113   from transformers import AutoModelForCausalLM, AutoTokenizer
114   from utils import tokenize_prompt_and_output, get_response_log_probs
115  -from sample import load_prompt_template, load_dataset_with_prompt_template,
         sample_rollout
116  +from sample import load_prompt_template, load_dataset_with_prompt_template,
         sample_rollout, MathContextBuffer, extract_mathematical_expressions,
         create_context_prompt
117   from drgrpo_grader import r1_zero_reward_fn_train
118   from evaluate import r1_zero_reward_fn_eval, evaluate_vllm
119   from grpo_utils import compute_group_normalized_rewards,
         grpo_microbatch_train_step
120  @@ -66,6 +66,9 @@ def evaluate_model(policy_model, vllm_model, eval_prompts,
         eval_answers, eval_s
121   def train_loop(model, train_prompts, train_answers, learning_rate, grpo_steps,
         train_steps_per_rollout, output_dir, batch_size, gradient_accumulation_steps =
          4, group_size = 2, rollout_subset_size = 256, device = "cuda", logging_steps
         = 20, saving_steps = 4000, eval_epochs = 5, eval_prompts = None, eval_answers
         = None, sampling_params = None, eval_vllm_model = None, cliprange = 0.2,
         loss_type = "reinforce_with_baseline"):
122       model.to(device)
123       training_steps = grpo_steps
124  +
125  +     # Initialize mathematical context buffer
126  +     context_buffer = MathContextBuffer(max_size=100)
127       optimizer = torch.optim.AdamW(model.parameters(), lr=learning_rate,
         weight_decay=0.0, betas=(0.9, 0.95))
128       global_step = 0  # Initialize global step counter
129
130  @@ -85,8 +88,31 @@ def train_loop(model, train_prompts, train_answers,
         learning_rate, grpo_steps,
131           load_policy_into_vllm_instance(model, vllm_model)
132
133           ## sample rollouts
134  +         # Get enhanced prompts with mathematical context for this epoch
135           print ("Sampling rollouts for epoch: ", epoch)
136  -         rollout_prompts, rollout_answers, rollout_responses, rollout_rewards =
         sample_rollout(vllm_model, r1_zero_reward_fn_train, train_prompts,
         train_answers, G=group_size, eval_sampling_params=eval_sampling_params,
         subset_size=rollout_subset_size, return_rewards=True, batch_size=512)
137  +
138  +         # Create enhanced prompts with context for this batch
139  +         enhanced_prompts = []
```

```
140 +          base_template = load_prompt_template()
141 +
142 +          # Select subset of problems for this epoch
143 +          if rollout_subset_size is not None:
144 +              indices = random.sample(range(len(train_prompts)),
        rollout_subset_size)
145 +              epoch_prompts = [train_prompts[i] for i in indices]
146 +              epoch_answers = [train_answers[i] for i in indices]
147 +          else:
148 +              epoch_prompts = train_prompts
149 +              epoch_answers = train_answers
150 +
151 +          # Create context-enhanced prompts
152 +          for prompt in epoch_prompts:
153 +              relevant_context = context_buffer.get_relevant_context(prompt,
        max_items=8)
154 +              enhanced_prompt = create_context_prompt(base_template.format(question
        =prompt.split("Question: ")[-1]), relevant_context)
155 +              enhanced_prompts.append(enhanced_prompt)
156 +
157 +          # Sample with enhanced prompts (need to adapt sample_rollout for direct
        prompt input)
158 +          rollout_prompts, rollout_answers, rollout_responses, rollout_rewards =
        sample_rollout_with_prompts(vllm_model, r1_zero_reward_fn_train,
        enhanced_prompts, epoch_answers, G=group_size, eval_sampling_params=
        eval_sampling_params, return_rewards=True, batch_size=512)
159 +
160          # Randomly sample 2 rollouts to print
161          indices = random.sample(range(len(rollout_prompts)), 2)
162          print ("Example rollouts:")
163 @@ -96,6 +122,15 @@ def train_loop(model, train_prompts, train_answers,
        learning_rate, grpo_steps,
164              print(f"Response: {rollout_responses[idx]}")
165              print(f"Reward: {rollout_rewards[idx]}")
166              print(f"Ground truth: {rollout_answers[idx]}")
167 +
168 +          # Update context buffer with mathematical expressions from responses
169 +          print("Updating mathematical context buffer...")
170 +          for response in rollout_responses:
171 +              if rollout_rewards[rollout_responses.index(response)] > 0:  # Only
        add from correct responses
172 +                  mathematical_expressions = extract_mathematical_expressions(
        response)
173 +                  if mathematical_expressions:
174 +                      context_buffer.add_expressions(mathematical_expressions)
175 +
176          rollout_tokenized = tokenize_prompt_and_output(rollout_prompts,
        rollout_responses, tokenizer)
177          rollout_data_loader = create_data_loader(rollout_tokenized, batch_size=
        batch_size, shuffle=False)
178
179 @@ -196,6 +231,47 @@ def train_loop(model, train_prompts, train_answers,
        learning_rate, grpo_steps,
180                  }, step=global_step)
181                  # print ("Global Step: ", global_step, "Loss: ", loss.item(),
         "Entropy: ", entropy.mean().item(), "Clip fraction: ", metadata.get("
        clip_fraction", 0.0))
182
183 +def sample_rollout_with_prompts(
184 +    vllm_model,
185 +    reward_fn,
186 +    prompts,
187 +    answers,
```

```
188  +      G,
189  +      eval_sampling_params,
190  +      return_rewards=False,
191  +      batch_size=64
192  +):
193  +      """Sample rollouts using provided prompts directly."""
194  +      # Create batched prompts by repeating each prompt G times
195  +      batched_prompts = []
196  +      batched_answers = []
197  +      for prompt, answer in zip(prompts, answers):
198  +          batched_prompts.extend([prompt] * G)
199  +          batched_answers.extend([answer] * G)
200  +
201  +      # Process in batches to avoid OOM
202  +      all_outputs = []
203  +      for i in range(0, len(batched_prompts), batch_size):
204  +          batch_prompts = batched_prompts[i:i + batch_size]
205  +          batch_outputs = vllm_model.generate(batch_prompts, eval_sampling_params)
206  +          all_outputs.extend(batch_outputs)
207  +
208  +      # Process results
209  +      batched_responses = []
210  +      batched_rewards = []
211  +      total_rewards = 0
212  +
213  +      for output, answer in tqdm(zip(all_outputs, batched_answers)):
214  +          generated_answer = output.outputs[0].text
215  +          reward = reward_fn(generated_answer, answer)
216  +          batched_responses.append(generated_answer)
217  +          batched_rewards.append(reward["reward"])
218  +          total_rewards += reward["reward"]
219  +
220  +      print(f"Accuracy of sampled rollouts: {total_rewards}/{len(batched_prompts)}
        = {total_rewards / len(batched_prompts) * 100}%")
221  +
222  +      return batched_prompts, batched_answers, batched_responses, batched_rewards
223  +
224   if __name__ == "__main__":
225       parser = argparse.ArgumentParser()
226       parser.add_argument("--model_path", type=str, default="Qwen/Qwen2.5-Math-1.5B
         ")
227  @@ -203,7 +279,7 @@ if __name__ == "__main__":
228       parser.add_argument("--train_dataset_path", type=str, default="../MATH/train.
         jsonl")
229       parser.add_argument("--eval_dataset_path", type=str, default="../MATH/test.
         jsonl")
230       parser.add_argument("--output_dir", type=str, default="ckpts/")
231  -     parser.add_argument("--learning_rate", type=float, default=1e-5)
232  +     parser.add_argument("--learning_rate", type=float, default=2.5e-5)
233       parser.add_argument("--grpo_steps", type=int, default=200)
234       parser.add_argument("--group_size", type=int, default=8)
235       parser.add_argument("--rollout_subset_size", type=int, default=256)
236  @@ -212,7 +288,7 @@ if __name__ == "__main__":
237       parser.add_argument("--gradient_accumulation_steps", type=int, default=16)
238       parser.add_argument("--batch_size", type=int, default=4)
239       parser.add_argument("--cliprange", type=float, default=0.2)
240  -     parser.add_argument("--loss_type", type=str, default="grpo_clip")
241  +     parser.add_argument("--loss_type", type=str, default="reinforce_with_baseline
         ")
242       parser.add_argument("--wandb_project", type=str, default="grpo-math")
243       parser.add_argument("--wandb_name", type=str, default="grpo_clip_1")
244       args = parser.parse_args()
245  --- repo_variants_full_pipeline_bsz50_fixed2_epoch7/idea_43/run_job.sh
```

```
246  +++ repo_variants_full_pipeline_bsz50_fixed2_epoch7/idea_43/run_job.sh
247  @@ -21,7 +21,7 @@ timeout 2h uv run    \
248        --index https://download.pytorch.org/whl/cu128 \
249        --index-strategy unsafe-best-match \
250      python grpo.py \
251  -        --learning_rate 1e-5 \
252  +        --learning_rate 2.5e-5 \
253          --grpo_steps 20 \
254          --group_size 8 \
255          --rollout_subset_size 128 \
256  @@ -30,7 +30,7 @@ timeout 2h uv run    \
257          --gradient_accumulation_steps 16 \
258          --batch_size 4 \
259          --cliprange 0.2 \
260  -        --loss_type grpo_clip \
261  +        --loss_type reinforce_with_baseline \
262          --wandb_name $wandb_name
263
264   echo "Experiment finished successfully!"
```

**Example 2**

**[Experiment]** Implement mathematical solution robustness training through systematic perturbation testing that teaches the model to solve mathematically equivalent problems with varied presentations, notation styles, and problem phrasings. Combine this with proven reward shaping by creating robustness-aware rewards that encourage mathematical understanding that generalizes across problem variations.

**[Code Changes]** Create problem perturbation system in `sample.py` that generates equivalent mathematical problems with varied notation, different variable names, alternative problem phrasing, and equivalent mathematical formulations. Track solution consistency across perturbations. Enhance `r1_zero_reward_fn_train` in `drgrpo_grader.py` to reward robustness: `robustness_bonus = 0.12 * consistency_score_across_perturbations` where consistency measures solution method stability across equivalent problems. Apply generalization bonuses: `+0.08` for solving problems with unfamiliar notation, `+0.06` for handling varied problem presentations. Modify `train_loop` in `grpo.py` to include perturbation-based training batches and track robustness improvement. Use robustness-aware learning rate with cosine annealing from `3.4e-5` to `1.8e-5`. Implement perturbation-complexity gradient clipping. Set `–loss_type reinforce_with_baseline`.

**Validation Accuracy: 56.2**

---

**Example 2 code diff**

```
1  --- repo_variants_full_pipeline_bsz50_fixed2_epoch7/idea_37/sample.py
2  +++ repo_variants_full_pipeline_bsz50_fixed2_epoch7/idea_37/sample.py
3  @@ -2,6 +2,7 @@ from vllm import LLM, SamplingParams
4   import json
5   from tqdm import tqdm
6   import random
7  +import re
8
9   def load_prompt_template(prompt_path = "prompts/r1_zero.prompt"):
10      with open(prompt_path, 'r') as f:
11  @@ -29,6 +30,125 @@ def load_dataset_with_prompt_template(prompt_template,
       dataset_path="/juice5b/s
12      answers = [example["answer"] for example in dataset]
13      return prompts, answers
14
15  +def create_notation_perturbations(problem_text):
16  +    """Create mathematical notation variations"""
17  +    perturbations = []
18  +
19  +    # Variable substitutions
20  +    var_substitutions = [
21  +        ('x', 'a'), ('y', 'b'), ('z', 'c'),
22  +        ('n', 'm'), ('k', 'j'), ('p', 'q'),
23  +        ('r', 's'), ('t', 'u'), ('v', 'w')
24  +    ]
25  +
26  +    # Create variable name variations
27  +    for old_var, new_var in var_substitutions:
28  +        if old_var in problem_text:
29  +            new_text = problem_text.replace(old_var, new_var)
30  +            if new_text != problem_text:
31  +                perturbations.append(("notation_var", new_text))
32  +
33  +    # Mathematical symbol variations
34  +    symbol_substitutions = [
35  +        ('\\cdot', '\\times'),
36  +        ('\\times', '\\cdot'),
37  +        ('\\frac', '\\div'),
```

```
38 +            ('\\le', '\\leq'),
39 +            ('\\ge', '\\geq'),
40 +        ]
41 +
42 +    for old_sym, new_sym in symbol_substitutions:
43 +        if old_sym in problem_text:
44 +            new_text = problem_text.replace(old_sym, new_sym)
45 +            perturbations.append(("notation_symbol", new_text))
46 +
47 +    return perturbations
48 +
49 +def create_phrasing_perturbations(problem_text):
50 +    """Create alternative problem phrasings"""
51 +    perturbations = []
52 +
53 +    # Common phrasing substitutions
54 +    phrasing_patterns = [
55 +        (r"Find the value of", "Determine"),
56 +        (r"What is", "Calculate"),
57 +        (r"Solve for", "Find"),
58 +        (r"How many", "What is the number of"),
59 +        (r"Compute", "Find"),
60 +        (r"Evaluate", "Calculate"),
61 +    ]
62 +
63 +    for pattern, replacement in phrasing_patterns:
64 +        if re.search(pattern, problem_text, re.IGNORECASE):
65 +            new_text = re.sub(pattern, replacement, problem_text, flags=re.
    IGNORECASE)
66 +            if new_text != problem_text:
67 +                perturbations.append(("phrasing", new_text))
68 +
69 +    return perturbations
70 +
71 +def create_formulation_perturbations(problem_text):
72 +    """Create equivalent mathematical formulations"""
73 +    perturbations = []
74 +
75 +    # Simple algebraic reformulations
76 +    reformulations = [
77 +        (r"(\w+) = (\d+) \+ (\d+)", r"\1 - \2 = \3"),  # a = b + c -> a - b = c
78 +        (r"(\w+) \+ (\d+) = (\d+)", r"\1 = \3 - \2"),  # x + a = b -> x = b - a
79 +        (r"(\d+) - (\w+) = (\d+)", r"\2 = \1 - \3"),  # a - x = b -> x = a - b
80 +    ]
81 +
82 +    for pattern, replacement in reformulations:
83 +        if re.search(pattern, problem_text):
84 +            new_text = re.sub(pattern, replacement, problem_text)
85 +            if new_text != problem_text:
86 +                perturbations.append(("formulation", new_text))
87 +
88 +    return perturbations
89 +
90 +def generate_problem_perturbations(problem_text, max_perturbations=3):
91 +    """Generate various perturbations of a mathematical problem"""
92 +    all_perturbations = []
93 +
94 +    # Generate different types of perturbations
95 +    all_perturbations.extend(create_notation_perturbations(problem_text))
96 +    all_perturbations.extend(create_phrasing_perturbations(problem_text))
97 +    all_perturbations.extend(create_formulation_perturbations(problem_text))
98 +
99 +    # Randomly sample up to max_perturbations
```

```
100 +    if len(all_perturbations) > max_perturbations:
101 +        all_perturbations = random.sample(all_perturbations, max_perturbations)
102 +
103 +    return all_perturbations
104 +
105 +def sample_with_perturbations(prompts, answers, perturbation_ratio=0.3):
106 +    """Sample problems with perturbations mixed in"""
107 +    perturbed_prompts = []
108 +    perturbed_answers = []
109 +    perturbation_metadata = []
110 +
111 +    for prompt, answer in zip(prompts, answers):
112 +        # Always include original
113 +        perturbed_prompts.append(prompt)
114 +        perturbed_answers.append(answer)
115 +        perturbation_metadata.append({"type": "original", "source_idx": len(
        perturbed_prompts)-1})
116 +
117 +        # Add perturbations with some probability
118 +        if random.random() < perturbation_ratio:
119 +            # Extract problem text from prompt (assuming it's in a specific
        format)
120 +            problem_text = prompt.split("question:")[-1] if "question:" in prompt
         else prompt
121 +            perturbations = generate_problem_perturbations(problem_text)
122 +
123 +            for pert_type, pert_text in perturbations:
124 +                # Reconstruct full prompt with perturbed problem
125 +                pert_prompt = prompt.replace(problem_text, pert_text)
126 +                perturbed_prompts.append(pert_prompt)
127 +                perturbed_answers.append(answer)  # Same answer for equivalent
        problem
128 +                perturbation_metadata.append({
129 +                    "type": pert_type,
130 +                    "source_idx": len(perturbed_prompts)-len(perturbations)-1
131 +                })
132 +
133 +    return perturbed_prompts, perturbed_answers, perturbation_metadata
134
135  def sample_rollout(
136      vllm_model,
137 --- repo_variants_full_pipeline_bsz50_fixed2_epoch7/idea_37/drgrpo_grader.py
138 +++ repo_variants_full_pipeline_bsz50_fixed2_epoch7/idea_37/drgrpo_grader.py
139 @@ -1025,3 +1025,83 @@ def r1_zero_reward_fn_train(response, ground_truth, fast=
        True):
140                "reward": 0.0
141            }
142
143 +def compute_consistency_score(responses, ground_truth, perturbation_metadata):
144 +    """Compute consistency score across perturbations"""
145 +    if not perturbation_metadata:
146 +        return 0.0
147 +
148 +    # Group responses by source problem
149 +    source_groups = {}
150 +    for i, meta in enumerate(perturbation_metadata):
151 +        source_idx = meta.get("source_idx", i)
152 +        if source_idx not in source_groups:
153 +            source_groups[source_idx] = []
154 +        source_groups[source_idx].append((i, responses[i], meta))
155 +
156 +    total_consistency = 0.0
157 +    valid_groups = 0
```

```
158  +
159  +    for source_idx, group_data in source_groups.items():
160  +        if len(group_data) > 1:  # Only consider groups with multiple responses
161  +            # Get correctness for each response in the group
162  +            correctness_scores = []
163  +            for _, response, meta in group_data:
164  +                reward_result = r1_zero_reward_fn_train(response, ground_truth)
165  +                correctness_scores.append(reward_result["reward"])
166  +
167  +            # Consistency is measured as how often the model gets the same
     correctness
168  +            if correctness_scores:
169  +                # If all responses have same correctness (all correct or all
     incorrect), high consistency
170  +                unique_scores = set(correctness_scores)
171  +                if len(unique_scores) == 1:
172  +                    consistency = 1.0
173  +                else:
174  +                    # Partial consistency based on agreement
175  +                    most_common = max(set(correctness_scores), key=
     correctness_scores.count)
176  +                    consistency = correctness_scores.count(most_common) / len(
     correctness_scores)
177  +
178  +                total_consistency += consistency
179  +                valid_groups += 1
180  +
181  +    if valid_groups == 0:
182  +        return 0.0
183  +
184  +    return total_consistency / valid_groups
185  +
186  +def r1_zero_reward_fn_train_robust(response, ground_truth, perturbation_metadata=
     None, responses_batch=None, fast=True):
187  +    """Enhanced reward function with robustness bonuses"""
188  +    # Get base reward
189  +    base_reward = r1_zero_reward_fn_train(response, ground_truth, fast)
190  +
191  +    # Initialize bonus components
192  +    robustness_bonus = 0.0
193  +    notation_bonus = 0.0
194  +    presentation_bonus = 0.0
195  +
196  +    # Compute robustness bonus if perturbation data available
197  +    if perturbation_metadata and responses_batch:
198  +        consistency_score = compute_consistency_score(responses_batch,
     ground_truth, perturbation_metadata)
199  +        robustness_bonus = 0.12 * consistency_score
200  +
201  +    # Check for notation and presentation bonuses from metadata
202  +    if perturbation_metadata:
203  +        current_meta = perturbation_metadata[0] if perturbation_metadata else {}
204  +        pert_type = current_meta.get("type", "original")
205  +
206  +        if pert_type in ["notation_var", "notation_symbol"] and base_reward["
     reward"] > 0:
207  +            notation_bonus = 0.08
208  +
209  +        if pert_type in ["phrasing", "formulation"] and base_reward["reward"] >
     0:
210  +            presentation_bonus = 0.06
211  +
212  +    # Combine all rewards
```

```
213 +    total_reward = base_reward["reward"] + robustness_bonus + notation_bonus +
         presentation_bonus
214 +
215 +    return {
216 +        "format_reward": base_reward["format_reward"],
217 +        "answer_reward": base_reward["answer_reward"],
218 +        "robustness_bonus": robustness_bonus,
219 +        "notation_bonus": notation_bonus,
220 +        "presentation_bonus": presentation_bonus,
221 +        "reward": total_reward
222 +    }
223 --- repo_variants_full_pipeline_bsz50_fixed2_epoch7/idea_37/grpo.py
224 +++ repo_variants_full_pipeline_bsz50_fixed2_epoch7/idea_37/grpo.py
225 @@ -1,7 +1,7 @@
226  from transformers import AutoModelForCausalLM, AutoTokenizer
227  from utils import tokenize_prompt_and_output, get_response_log_probs
228 -from sample import load_prompt_template, load_dataset_with_prompt_template,
         sample_rollout
229 -from drgrpo_grader import r1_zero_reward_fn_train
230 +from sample import load_prompt_template, load_dataset_with_prompt_template,
         sample_rollout, sample_with_perturbations
231 +from drgrpo_grader import r1_zero_reward_fn_train, r1_zero_reward_fn_train_robust
232  from evaluate import r1_zero_reward_fn_eval, evaluate_vllm
233  from grpo_utils import compute_group_normalized_rewards,
         grpo_microbatch_train_step
234  from torch.utils.data import DataLoader, Dataset
235 @@ -12,6 +12,7 @@ from tqdm import tqdm
236  from vllm import LLM, SamplingParams
237  import wandb
238  import random
239 +import math
240
241  def load_policy_into_vllm_instance(policy, llm):
242      state_dict = policy.state_dict()
243 @@ -63,11 +64,23 @@
244      metrics = evaluate_vllm(vllm_model, r1_zero_reward_fn_eval, eval_prompts,
         eval_answers, eval_sampling_params, output_path=output_path)
245      return metrics
246
247 -def train_loop(model, train_prompts, train_answers, learning_rate, grpo_steps,
         train_steps_per_rollout, output_dir, batch_size, gradient_accumulation_steps =
         4, group_size = 2, rollout_subset_size = 256, device = "cuda", logging_steps
         = 20, saving_steps = 4000, eval_epochs = 5, eval_prompts = None, eval_answers
         = None, sampling_params = None, eval_vllm_model = None, cliprange = 0.2,
         loss_type = "reinforce_with_baseline"):
248 +def get_cosine_annealing_lr(step, total_steps, lr_max=3.4e-5, lr_min=1.8e-5):
249 +    """Cosine annealing learning rate schedule"""
250 +    return lr_min + (lr_max - lr_min) * 0.5 * (1 + math.cos(math.pi * step /
         total_steps))
251 +
252 +def compute_perturbation_complexity_clip_norm(model, complexity_factor=1.0):
253 +    """Compute gradient clipping norm based on perturbation complexity"""
254 +    base_clip_norm = 1.0
255 +    return base_clip_norm * complexity_factor
256 +
257 +def train_loop(model, train_prompts, train_answers, learning_rate, grpo_steps,
         train_steps_per_rollout, output_dir, batch_size, gradient_accumulation_steps =
         4, group_size = 2, rollout_subset_size = 256, device = "cuda", logging_steps
         = 20, saving_steps = 4000, eval_epochs = 5, eval_prompts = None, eval_answers
         = None, sampling_params = None, eval_vllm_model = None, cliprange = 0.2,
         loss_type = "reinforce_with_baseline", perturbation_ratio = 0.3):
258      model.to(device)
259      training_steps = grpo_steps
```

```
260  -     optimizer = torch.optim.AdamW(model.parameters(), lr=learning_rate,
           weight_decay=0.0, betas=(0.9, 0.95))
261  +     # Start with robustness-aware learning rate
262  +     initial_lr = get_cosine_annealing_lr(0, grpo_steps)
263  +     optimizer = torch.optim.AdamW(model.parameters(), lr=initial_lr, weight_decay
           =0.0, betas=(0.9, 0.95))
264        global_step = 0  # Initialize global step counter
265  +     robustness_scores = []
266
267        for epoch in range(grpo_steps):
268            model.train()
269  @@ -82,21 +95,50 @@ def train_loop(model, train_prompts, train_answers,
           learning_rate, grpo_steps,
270
271                model.train()
272
273  +            # Update learning rate with cosine annealing
274  +            current_lr = get_cosine_annealing_lr(epoch, grpo_steps)
275  +            for param_group in optimizer.param_groups:
276  +                param_group['lr'] = current_lr
277  +
278                ## load the current policy model to vllm for sampling rollouts
279                load_policy_into_vllm_instance(model, vllm_model)
280
281  +            # Sample with perturbations for robustness training
282  +            perturbed_prompts, perturbed_answers, perturbation_metadata =
           sample_with_perturbations(
283  +                train_prompts, train_answers, perturbation_ratio=perturbation_ratio
284  +            )
285  +
286                ## sample rollouts
287                print ("Sampling rollouts for epoch: ", epoch)
288  -            rollout_prompts, rollout_answers, rollout_responses, rollout_rewards =
           sample_rollout(vllm_model, r1_zero_reward_fn_train, train_prompts,
           train_answers, G=group_size, eval_sampling_params=eval_sampling_params,
           subset_size=rollout_subset_size, return_rewards=True, batch_size=512)
289  +
290  +            # Use subset of perturbed prompts for training
291  +            subset_size = min(rollout_subset_size, len(perturbed_prompts))
292  +            if subset_size < len(perturbed_prompts):
293  +                indices = random.sample(range(len(perturbed_prompts)), subset_size)
294  +                subset_prompts = [perturbed_prompts[i] for i in indices]
295  +                subset_answers = [perturbed_answers[i] for i in indices]
296  +                subset_metadata = [perturbation_metadata[i] for i in indices]
297  +            else:
298  +                subset_prompts = perturbed_prompts
299  +                subset_answers = perturbed_answers
300  +                subset_metadata = perturbation_metadata
301  +
302  +            rollout_prompts, rollout_answers, rollout_responses, rollout_rewards =
           sample_rollout(
303  +                vllm_model, r1_zero_reward_fn_train, subset_prompts, subset_answers,
304  +                G=group_size, eval_sampling_params=eval_sampling_params,
305  +                subset_size=None, return_rewards=True, batch_size=512
306  +            )
307  +
308                # Randomly sample 2 rollouts to print
309                indices = random.sample(range(len(rollout_prompts)), 2)
310                print ("Example rollouts:")
311                for idx in indices:
312                    print(f"\nRollout {idx}:")
313  -                print(f"Prompt: {rollout_prompts[idx]}")
314  +                print(f"Prompt: {rollout_prompts[idx][:200]}...")
```

```
315              print(f"Response: {rollout_responses[idx]}")
316              print(f"Reward: {rollout_rewards[idx]}")
317 -            print(f"Ground truth: {rollout_answers[idx]}")
318 +            print(f"Ground truth: {rollout_answers[idx][:100]}...")
319 +
320          rollout_tokenized = tokenize_prompt_and_output(rollout_prompts,
      rollout_responses, tokenizer)
321          rollout_data_loader = create_data_loader(rollout_tokenized, batch_size=
      batch_size, shuffle=False)
322
323 @@ -126,15 +168,32 @@ def train_loop(model, train_prompts, train_answers,
      learning_rate, grpo_steps,
324
325          # Compute advantages using group normalization - no gradients needed
326          with torch.no_grad():
327 -            advantages, raw_rewards, metadata = compute_group_normalized_rewards(
328 -                reward_fn=r1_zero_reward_fn_train,
329 +            # Create enhanced reward function with robustness
330 +            def robust_reward_fn(response, ground_truth):
331 +                # Find corresponding metadata for this response
332 +                response_idx = rollout_responses.index(response) if response in
      rollout_responses else 0
333 +                meta_idx = min(response_idx, len(subset_metadata) - 1)
334 +                current_meta = [subset_metadata[meta_idx]] if subset_metadata
      else None
335 +
336 +                return r1_zero_reward_fn_train_robust(
337 +                    response, ground_truth,
338 +                    perturbation_metadata=current_meta,
339 +                    responses_batch=rollout_responses
340 +                )
341 +
342 +            advantages, raw_rewards, metadata = compute_group_normalized_rewards(
343 +                reward_fn=robust_reward_fn,
344                  rollout_responses=rollout_responses,
345                  repeated_ground_truths=rollout_answers,
346                  group_size=group_size,
347                  advantage_eps=1e-6,
348                  normalize_by_std=True
349              )
350              advantages = advantages.to(device)
351 +
352 +            # Track robustness improvement
353 +            current_robustness = metadata.get('mean_reward', 0.0)
354 +            robustness_scores.append(current_robustness)
355
356          # Log raw rewards statistics
357          print("\nGRPO epoch: ", epoch)
358 @@ -145,11 +204,20 @@
359              wandb.log({
360                  "eval/mean_reward": eval_mean_reward,
361                  "train/mean_reward": metadata["mean_reward"],
362 +                "train/learning_rate": current_lr,
363 +                "train/robustness_score": current_robustness,
364              }, step=global_step)
365          else:
366              wandb.log({
367                  "train/mean_reward": metadata["mean_reward"],
368 +                "train/learning_rate": current_lr,
369 +                "train/robustness_score": current_robustness,
370              }, step=global_step)
371 +
372 +        # Compute perturbation complexity for gradient clipping
```

```
373 +           perturbation_types = set(meta.get("type", "original") for meta in
        subset_metadata)
374 +           complexity_factor = 1.0 + 0.1 * len(perturbation_types)  # More complex
        with more perturbation types
375 +           clip_norm = compute_perturbation_complexity_clip_norm(model,
        complexity_factor)
376
377
378           ## train on this rollout batch for train_steps_per_rollout steps
379 @@ -185,6 +252,9 @@ def train_loop(model, train_prompts, train_answers,
        learning_rate, grpo_steps,
380                   )
381
382                   if (batch_idx + 1) % gradient_accumulation_steps == 0:
383 +                       # Apply perturbation-complexity gradient clipping
384 +                       torch.nn.utils.clip_grad_norm_(model.parameters(), clip_norm)
385 +
386                       optimizer.step()
387                       optimizer.zero_grad()
388
389 @@ -211,7 +281,7 @@ if __name__ == "__main__":
390     parser.add_argument("--gradient_accumulation_steps", type=int, default=16)
391     parser.add_argument("--batch_size", type=int, default=4)
392     parser.add_argument("--cliprange", type=float, default=0.2)
393 -   parser.add_argument("--loss_type", type=str, default="grpo_clip")
394 +   parser.add_argument("--loss_type", type=str, default="reinforce_with_baseline
        ")
395     parser.add_argument("--wandb_project", type=str, default="grpo-math")
396     parser.add_argument("--wandb_name", type=str, default="grpo_clip_1")
397     args = parser.parse_args()
398 @@ -266,7 +336,8 @@ if __name__ == "__main__":
399         sampling_params=eval_sampling_params,
400         eval_vllm_model=vllm_model,
401         cliprange=args.cliprange,
402 -       loss_type=args.loss_type
403 +       loss_type=args.loss_type,
404 +       perturbation_ratio=0.3
405     )
```

## Example 3

**[Experiment]** Implement token-level reward attribution by using attention weights to identify which input tokens contributed most to correct answers, then amplifying the gradient updates for those tokens during policy gradient training.

**[Code Changes]** Modify `get_response_log_probs` in `utils.py` to also return attention weights from the last layer. In `grpo_microbatch_train_step`, compute token importance scores by averaging attention weights across heads, then multiply the policy gradient loss by `(1 + importance_score)` for tokens with high attention to mathematical content.

**Validation Accuracy: 45.2**

Example 3 code diff

```
1  --- repo_variants_full_pipeline_bsz50_fixed2_epoch1/idea_32/utils.py
2  +++ repo_variants_full_pipeline_bsz50_fixed2_epoch1/idea_32/utils.py
3  @@ -41,12 +41,13 @@ def compute_entropy(logits):
4         entropy = -torch.sum(probs * log_probs, dim=-1)
5         return entropy
6
7  -def get_response_log_probs(model, input_ids, labels, return_token_entropy=False,
       no_grad=True):
8  +def get_response_log_probs(model, input_ids, labels, return_token_entropy=False,
       return_attention=False, no_grad=True):
9        if no_grad:
10           with torch.no_grad():
11  -             outputs = model(input_ids, labels=labels)
12  +             outputs = model(input_ids, labels=labels, output_attentions=
       return_attention)
13               logits = outputs.logits # (batch_size, seq_len, vocab_size)
14               log_probs = torch.log_softmax(logits, dim=-1) # (batch_size, seq_len,
        vocab_size)
15  +             attentions = outputs.attentions if return_attention else None
16               # Get log probs of the actual label tokens
17               batch_size, seq_len = labels.shape # (batch_size, seq_len)
18               log_probs = torch.gather(log_probs, dim=-1, index=labels.unsqueeze
       (-1)).squeeze(-1)
19  @@ -55,8 +56,9 @@ def get_response_log_probs(model, input_ids, labels,
       return_token_entropy=False
20               else:
21                   entropy = None
22         else:
23  -         outputs = model(input_ids, labels=labels)
24  +         outputs = model(input_ids, labels=labels, output_attentions=
       return_attention)
25           logits = outputs.logits # (batch_size, seq_len, vocab_size)
26  +         attentions = outputs.attentions if return_attention else None
27           log_probs = torch.log_softmax(logits, dim=-1) # (batch_size, seq_len,
       vocab_size)
28           # Get log probs of the actual label tokens
29           batch_size, seq_len = labels.shape # (batch_size, seq_len)
30  @@ -65,9 +67,17 @@ def get_response_log_probs(model, input_ids, labels,
       return_token_entropy=False
31               entropy = compute_entropy(logits)
32           else:
33               entropy = None
34  -
35  -     return {
36  +
37  +     result = {
38           "log_probs": log_probs,
```

```
39            "token_entropy": entropy
40         }
41  +
42  +     if return_attention and attentions is not None:
43  +         # Return attention weights from the last layer, averaged across heads
44  +         last_layer_attention = attentions[-1]  # Shape: (batch_size, num_heads,
        seq_len, seq_len)
45  +         averaged_attention = last_layer_attention.mean(dim=1)  # Average across
        heads: (batch_size, seq_len, seq_len)
46  +         result["attention_weights"] = averaged_attention
47  +
48  +     return result
49
50  --- repo_variants_full_pipeline_bsz50_fixed2_epoch1/idea_32/grpo_utils.py
51  +++ repo_variants_full_pipeline_bsz50_fixed2_epoch1/idea_32/grpo_utils.py
52  @@ -170,6 +170,7 @@ def grpo_microbatch_train_step(
53      advantages: torch.Tensor | None = None,
54      old_log_probs: torch.Tensor | None = None,
55      cliprange: float | None = None,
56  +   attention_weights: torch.Tensor | None = None,
57   ) -> tuple[torch.Tensor, dict[str, torch.Tensor]]:
58       '''
59       Return:
60  @@ -180,6 +181,20 @@ def grpo_microbatch_train_step(
61       You should call loss.backward() in this function. Make sure to adjust for
        gradient accumulation.
62       '''
63       loss, metadata = compute_policy_gradient_loss(policy_log_probs, loss_type,
        raw_rewards, advantages, old_log_probs, cliprange) # (batch_size,
        sequence_length)
64  +
65  +     # Apply attention-based weighting if attention weights are provided
66  +     if attention_weights is not None:
67  +         # Compute importance scores by averaging attention weights to earlier
        tokens
68  +         # attention_weights shape: (batch_size, seq_len, seq_len)
69  +         # For each token, sum attention weights to all previous tokens (including
         itself)
70  +         importance_scores = torch.sum(attention_weights, dim=-1)  # (batch_size,
        seq_len)
71  +
72  +         # Normalize importance scores to [0, 1] range per sequence
73  +         importance_scores = importance_scores / (importance_scores.max(dim=-1,
        keepdim=True)[0] + 1e-8)
74  +
75  +         # Amplify loss for tokens with high importance: multiply by (1 +
        importance_score)
76  +         loss = loss * (1.0 + importance_scores)
77  +
78       loss = masked_mean(loss, response_mask)
79       loss = loss / gradient_accumulation_steps
80       loss.backward()
81  --- repo_variants_full_pipeline_bsz50_fixed2_epoch1/idea_32/grpo.py
82  +++ repo_variants_full_pipeline_bsz50_fixed2_epoch1/idea_32/grpo.py
83  @@ -109,6 +109,7 @@ def train_loop(model, train_prompts, train_answers,
        learning_rate, grpo_steps,
84                     model,
85                     input_ids,
86                     labels,
87                     return_token_entropy=False,
88  +                  return_attention=False,
89                     no_grad=True
90                 )
```

```
91   @@ -163,11 +164,13 @@
92                       model,
93                       input_ids,
94                       labels,
95                       return_token_entropy=True,
96   +                   return_attention=True,
97                       no_grad=False
98                   )
99                   policy_log_probs = response_log_probs["log_probs"]
100                  entropy = response_log_probs["token_entropy"]
101  +               attention_weights = response_log_probs.get("attention_weights")
102
103                  # Calculate data index for advantages/old_log_probs
104                  batch_idx_total = batch_idx * batch_size
105  @@ -177,7 +180,8 @@ def train_loop(model, train_prompts, train_answers,
         learning_rate, grpo_steps,
106                      loss_type=loss_type,
107                      advantages=batch_advantages,
108                      old_log_probs=batch_old_log_probs,
109  -                   cliprange=cliprange
110  +                   cliprange=cliprange,
111  +                   attention_weights=attention_weights
112                  )
113
114                  if (batch_idx + 1) % gradient_accumulation_steps == 0:
```

**Example 4**

**[Experiment]** Implement response diversity rewards within groups where responses to the same prompt receive bonus rewards (0.05-0.15) for being dissimilar to other responses in their group, encouraging exploration of different solution paths while maintaining the proven `group_size=8` and `3e-5` learning rate combination.

**[Code Changes]** Modify `compute_group_normalized_rewards` in `grpo_utils.py` to compute pairwise similarity between responses in each group using token-level Jaccard similarity. Add diversity bonus: `diversity_reward = 0.15 * (1 - max_similarity_in_group)` to each response's reward before advantage computation. Reshape responses into groups, compute similarities, and add bonuses before advantage normalization. Set `-learning_rate 3e-5`, `-loss_type reinforce_with_baseline`, `-group_size 8`.

**Validation Accuracy: 19.2**

```
Example 4 code diff

1  --- repo_variants_full_pipeline_bsz50_fixed2_epoch3/idea_22/grpo_utils.py
2  +++ repo_variants_full_pipeline_bsz50_fixed2_epoch3/idea_22/grpo_utils.py
3  @@ -1,5 +1,25 @@
4   import torch
5   from typing import Literal
6
7  +def compute_jaccard_similarity(response1: str, response2: str) -> float:
8  +    """
9  +    Compute token-level Jaccard similarity between two responses.
10 +    Jaccard similarity = |intersection| / |union|
11 +    """
12 +    # Simple tokenization by splitting on whitespace
13 +    tokens1 = set(response1.split())
14 +    tokens2 = set(response2.split())
15 +
16 +    if len(tokens1) == 0 and len(tokens2) == 0:
17 +        return 1.0
18 +
19 +    intersection = tokens1.intersection(tokens2)
20 +    union = tokens1.union(tokens2)
21 +
22 +    if len(union) == 0:
23 +        return 1.0
24 +
25 +    return len(intersection) / len(union)
26 +
27  def compute_group_normalized_rewards(
28      reward_fn,
29 @@ -30,31 +49,62 @@
30          raw_rewards.append(reward_dict["reward"])
31      raw_rewards = torch.tensor(raw_rewards)
32
33 +    # Compute diversity bonuses within each group
34 +    n_groups = len(rollout_responses) // group_size
35 +    diversity_bonuses = []
36 +
37 +    for group_idx in range(n_groups):
38 +        group_start = group_idx * group_size
39 +        group_end = group_start + group_size
40 +        group_responses = rollout_responses[group_start:group_end]
41 +
42 +        # Compute pairwise similarities within the group
43 +        for i in range(group_size):
44 +            max_similarity = 0.0
45 +            for j in range(group_size):
```

```
46  +                    if i != j:
47  +                        similarity = compute_jaccard_similarity(group_responses[i],
        group_responses[j])
48  +                        max_similarity = max(max_similarity, similarity)
49  +
50  +                # Diversity bonus: higher reward for more dissimilar responses
51  +                diversity_bonus = 0.15 * (1 - max_similarity)
52  +                diversity_bonuses.append(diversity_bonus)
53  +
54  +        diversity_bonuses = torch.tensor(diversity_bonuses)
55  +
56          # Reshape rewards into groups
57          n_groups = len(raw_rewards) // group_size
58          grouped_rewards = raw_rewards.view(n_groups, group_size)
59  +        grouped_diversity_bonuses = diversity_bonuses.view(n_groups, group_size)
60  +
61  +        # Add diversity bonuses to raw rewards before advantage computation
62  +        grouped_rewards_with_diversity = grouped_rewards + grouped_diversity_bonuses
63
64          # Compute group statistics
65  -        group_means = grouped_rewards.mean(dim=1, keepdim=True)
66  +        group_means = grouped_rewards_with_diversity.mean(dim=1, keepdim=True)
67          if normalize_by_std:
68  -            group_stds = grouped_rewards.std(dim=1, keepdim=True) + advantage_eps
69  -            advantages = (grouped_rewards - group_means) / group_stds
70  +            group_stds = grouped_rewards_with_diversity.std(dim=1, keepdim=True) +
        advantage_eps
71  +            advantages = (grouped_rewards_with_diversity - group_means) / group_stds
72          else:
73  -            advantages = grouped_rewards - group_means
74  +            advantages = grouped_rewards_with_diversity - group_means
75
76          # Flatten advantages back to original shape
77          advantages = advantages.view(-1)
78  +
79  +        # Update raw rewards to include diversity bonuses for metadata
80  +        raw_rewards_with_diversity = raw_rewards + diversity_bonuses
81
82          # Compute metadata statistics
83          metadata = {
84  -            "mean_reward": raw_rewards.mean().item(),
85  -            "std_reward": raw_rewards.std().item(),
86  -            "max_reward": raw_rewards.max().item(),
87  -            "min_reward": raw_rewards.min().item(),
88  +            "mean_reward": raw_rewards_with_diversity.mean().item(),
89  +            "std_reward": raw_rewards_with_diversity.std().item(),
90  +            "max_reward": raw_rewards_with_diversity.max().item(),
91  +            "min_reward": raw_rewards_with_diversity.min().item(),
92              "mean_advantage": advantages.mean().item(),
93              "std_advantage": advantages.std().item(),
94  +            "mean_diversity_bonus": diversity_bonuses.mean().item(),
95          }
96
97  -    return advantages, raw_rewards, metadata
98  +    return advantages, raw_rewards_with_diversity, metadata
99
100  def compute_naive_policy_gradient_loss(
101 --- repo_variants_full_pipeline_bsz50_fixed2_epoch3/idea_22/grpo.py
102 +++ repo_variants_full_pipeline_bsz50_fixed2_epoch3/idea_22/grpo.py
103 @@ -203,6 +203,6 @@
104     parser.add_argument("--eval_dataset_path", type=str, default="../MATH/test.
        jsonl")
105     parser.add_argument("--output_dir", type=str, default="ckpts/")
```

```
106  -      parser.add_argument("--learning_rate", type=float, default=1e-5)
107  +      parser.add_argument("--learning_rate", type=float, default=3e-5)
108         parser.add_argument("--grpo_steps", type=int, default=200)
109         parser.add_argument("--group_size", type=int, default=8)
110         parser.add_argument("--rollout_subset_size", type=int, default=256)
111  @@ -212,7 +212,7 @@ if __name__ == "__main__":
112         parser.add_argument("--gradient_accumulation_steps", type=int, default=16)
113         parser.add_argument("--batch_size", type=int, default=4)
114         parser.add_argument("--cliprange", type=float, default=0.2)
115  -      parser.add_argument("--loss_type", type=str, default="grpo_clip")
116  +      parser.add_argument("--loss_type", type=str, default="reinforce_with_baseline
     ")
117         parser.add_argument("--wandb_project", type=str, default="grpo-math")
118         parser.add_argument("--wandb_name", type=str, default="grpo_clip_1")
119         args = parser.parse_args()
120  --- repo_variants_full_pipeline_bsz50_fixed2_epoch3/idea_22/run_job.sh
121  +++ repo_variants_full_pipeline_bsz50_fixed2_epoch3/idea_22/run_job.sh
122  @@ -21,7 +21,7 @@ timeout 2h uv run    \
123         --index https://download.pytorch.org/whl/cu128 \
124         --index-strategy unsafe-best-match \
125       python grpo.py \
126  -        --learning_rate 1e-5 \
127  +        --learning_rate 3e-5 \
128         --grpo_steps 20 \
129         --group_size 8 \
130         --rollout_subset_size 128 \
131  @@ -30,7 +30,7 @@ timeout 2h uv run    \
132         --gradient_accumulation_steps 16 \
133         --batch_size 4 \
134         --cliprange 0.2 \
135  -        --loss_type grpo_clip \
136  +        --loss_type reinforce_with_baseline \
137         --wandb_name $wandb_name
138
139    echo "Experiment finished successfully!"
```

