# OpenReview forum: "Towards Execution-Grounded Automated AI Research"
_ICML.cc/2026/Conference — ICML 2026 regular_

### Official Review · Reviewer_xGMf · 2026-03-11

**Soundness:** 3
**Presentation:** 3
**Significance:** 3
**Originality:** 4
**Overall Recommendation:** 4
**Confidence:** 4

**Summary:**

The paper introduces an automated execution system that takes natural language research ideas, converts them to code implementations, runs experiments, and returns benchmark performance. Using this system, the authors study whether LLMs can improve their ideation through two learning paradigms: evolutionary search and reinforcement learning from execution feedback. They evaluate on two environments (nanoGPT pretraining and GRPO-based math reasoning) with frontier models including Claude-4.5-Opus, Claude-4.5-Sonnet, and GPT-5. The key findings are that execution-guided evolutionary search can discover ideas that outperform both baselines and human experts, but shows limited scaling beyond certain models. RL from execution rewards improves average performance but collapses in diversity and fails to improve the upper bound.

**Compliance With Llm Reviewing Policy:**

Affirmed.

**Final Justification:**

Thank you to the authors for the response and the transparency in the rebuttal. The paper addresses a popular and significant research question promptly, and demonstrates meaningful results, though it's also clear that the paper is a bit preliminary in the context of "AI for research“ and could be further polished and improved for higher impact if more time and efforts are allowed. Therefore, I retain my assessment of 4 - Weak Accept.

**Key Questions For Authors:**

In summary, the paper timely addresses a popular topic and offers valuable insights towards AI for research, despite several missing details (as outlined below) and a feeling that the paper is still preliminary (though it's totally understandable that large-scale extensive experiments are not realistic given the limited resources in academia, and it doesn't hinder the fact that the paper could be a foundational work for further exploration in AI for research). I give the current score in light of the nice and timely idea showcased in the paper, and I would be willing to increase my score to 5 or 6 if the following concerns are resolved or well explained.


**On generalization**

(1) Have you tested whether the best-performing ideas (e.g., the 69.4% GRPO solution, the 3.1407 loss nanoGPT solution) transfer to other datasets or model sizes? If I implemented that idea on a different math dataset (say, GSM8K) or a larger model (say, 7B instead of 1.5B), would I see similar gains? The paper only tests a very specific scope in AI research (particularly, pre-training and post-training of LLMs where the goal is almost solely benchmark score chasing). Can it generalize to the entire AI research pipeline (e.g. data, infrastructure, etc. are also part of AI research) or even broader research schemes (e.g. physics, math, engineering, etc.), and if so, how do you plan to evaluate it?

**On the RL signal**

(2) The RL experiments use a reward of 0 for failed executions. Have you tried more granular rewards based on where execution fails (patch failure vs. runtime crash vs. numerical instability)? The paper's own analysis shows that many failures are due to executor limitations rather than bad ideas. If the model could distinguish between "this idea couldn't be implemented" and "this idea executed but performed poorly," would the learning dynamics change?

**On search sensitivity**

(3) The evolutionary search combines exploitation (variants of good ideas) and exploration (novel ideas) with an annealing schedule. How sensitive are the results to this schedule? If you ran Claude-4.5-Sonnet with a slower annealing rate, would it escape the early saturation you observe? The claim that some models "saturate early" might be a property of the search algorithm, not the model.

**On thinking length**

(4) The thinking length collapse in RL is striking. Do you have examples of long-thinking ideas that did execute successfully? What distinguished them from the ones that failed? Is it possible to predict which complex ideas are likely to be implementable, and could that prediction be used as part of the reward?

**On interpretability**

(5) AI research is not just about a marginal increase on a specific benchmark which is not a robust signal of improvement and possibly only due to some randomness. For the new algorithms discovered by AI, how is the interpretability of the gain in score? And is it possible that a top-tier peer-reviewed research paper could be produced by AI without human intervention (or with minimal human intervention)?

**Limitations:**

See "Weaknesses" and "Key Questions for Authors".

**Strengths And Weaknesses:**

**Strengths**

There are several strengths worth highlighting. First, the automated executor system itself is thoughtfully designed. The three-component architecture (Implementer, Scheduler, Worker) with patch generation, self-revision, and execution logging addresses real engineering challenges. The fact that they caught and prevented reward hacking (models leaking future tokens) during development shows careful attention to experimental validity.

The empirical results reveal genuinely interesting patterns. The discovery that Claude-4.5-Sonnet found vanilla policy gradient with group-average baseline outperforming the standard GRPO objective (Section 4.2) is the kind of serendipitous finding that makes this line of work compelling. The 19.7 minute training time on nanoGPT beating the 35.9 minute baseline is non-trivial—anyone who's optimized training runs knows how easy it is to make things worse.

The analysis distinguishing algorithmic ideas from hyperparameter tuning (Table 2) is valuable. The field often talks about "new algorithms" when we're really just turning knobs, so seeing that Claude-4.5-Opus generated 96.3% algorithmic ideas in GRPO (even if many failed) suggests these models aren't just memorizing hyperparameter grids. The diversity collapse analysis in RL (Section 5.3) is also illuminating, showing that longer thinking traces correlate with lower execution rates, leading the model to prefer shorter thinking, is the kind of mechanistic insight that points toward concrete fixes.

**Weaknesses**

**The generalization problem and overclaim with the wording "research".** This is the elephant in the room, and while the authors acknowledge it in Appendix A.2, I think it deserves more weight in the main text. They're optimizing for validation accuracy on MATH (a dataset that's been in the training data of every model they use) and for reaching a specific loss on FineWeb. When Claude-4.5-Sonnet discovers that vanilla policy gradient works better, is that a genuine algorithmic insight or overfitting to the specific reward structure of this environment? The paper doesn't test whether these ideas transfer to different datasets, different model sizes, or even different random seeds. Research that only works in one controlled environment isn't really research, instead it's hyperparameter optimization with extra steps.

**Distinguishing failure due to idea or implementation.** This came up repeatedly in my reading of the RL experiments. When an idea gets reward 0 because execution failed, the model learns nothing about why. Was the idea fundamentally flawed? Was the implementation buggy? Did it try to import a library that wasn't available? The paper's analysis of failures (Appendix A.4) shows that many failed ideas were actually quite sophisticated where they just required capabilities the executor didn't have. This means the RL signal is extremely noisy, and I'm not convinced the "average reward increases but max reward doesn't" finding (Section 5.2) is robust to better execution. If you had a perfect executor, would RL actually improve the upper bound? The paper can't answer this, but it's the question I care about most.

**The baseline comparison.** Table 1 compares against "Best Human Expert" but the contexts are very different. For GRPO, they compare to a graduate class assignment leaderboard. For nanoGPT, they compare to the speedrun leaderboard. These are real humans doing real work, so it's not invalid, but the humans weren't given 50-80 attempts per epoch over multiple epochs with automated execution. They wrote one solution and submitted it. If you gave humans the same budget of attempts (even without automation), I suspect the gap would narrow significantly. The comparison feels asymmetric and underspecified.

**The scaling claims.** The paper says "only Claude-4.5-Opus shows a clear scaling curve" (Section 4.2) but they only run 10 epochs. That's not enough to claim scaling or saturation. GPT-5 might look like it saturates early because it finds a local optimum in epoch 2-3, but with more epochs and better exploration it could break through. The evolutionary search algorithm itself (Algorithm 1) has annealing schedules that might prematurely converge. I'd want to see longer runs or more systematic variation of the search hyperparameters before concluding anything about model capabilities.

---

> ### Author Rebuttal · Authors · 2026-03-31
>
> - *"The paper doesn't test whether these ideas transfer to different datasets, different model sizes, or even different random seeds."*
>
> We acknowledge this is a limitation of our work (we acknowledged it as the first point in A.2 of our draft). Our current pretraining and post-training experiments are conducted on small model sizes, and unfortunately running a single generated idea requires a full GPU training run, making it computationally prohibitive to systematically replicate experiments across multiple seeds, datasets, or model sizes within the scope of this paper. Nonetheless, we believe our work provides the first empirical evidence that LLMs can learn to improve idea effectiveness via an evolutionary search loop — establishing the foundation and motivating future research on the generalizability and scalability of these methods.
>
> - *"Distinguishing failure due to idea or implementation. If you had a perfect executor, would RL actually improve the upper bound? The paper can't answer this, but it's the question I care about most."*
>
> This is an excellent research question. Our current RL reward design does not disentangle these two sources of failure — a negative result could stem from a poor idea, a flawed or unfaithful implementation, or a mixture of these factors. Disentangling them cleanly is a non-trivial task that could be interesting for future work to explore.
>
> The hypothetical question of whether a perfect executor can result in a different RL learning curve is intriguing. It’s impossible to verify this hypothesis in our current setup, given that current coding agents are not able to execute all the proposed ideas perfectly, but we agree it’d be interesting for future work to explore what happens when we swap in increasingly capable execution agents in our RL loop, as pointed out in the discussion section of our paper (A.2).
>
> It is also unclear how much agency the executor should have -- for example, the ideator can produce a bad idea, but the executor can still get inspired by it and produce a somewhat different execution that ends up working great.
>
> - *"Table 1 compares against "Best Human Expert" but the contexts are very different. The comparison feels asymmetric and underspecified."*
>
> We appreciate the reviewer for flagging this asymmetry. Rather than claiming parity with human researchers in either environment, the comparison is meant to provide an intuitive sense of how well the system performs — situating our system's outputs relative to what humans with domain knowledge can achieve. We have clarified the human baselines in the paper (last paragraph in Sec 4.2), and we will add some additional explanation to make it explicit that these humans are operating under a very different budget constraint than our system.

---

> > ### Author Rebuttal · Reviewer_xGMf · 2026-04-01
> >
> > Thank you for the response and the transparency in your rebuttal. I believe I'm on the same page as the authors on both the strengths and areas for future improvement of the paper. I think the paper addresses a popular and significant research question promptly, and demonstrates meaningful results, though it's also clear that the paper is a bit preliminary in the context of "AI for research“ and could be further polished and improved for higher impact if more time and efforts are allowed. Therefore, I will retain my assessment of 4 - Weak Accept for now, but will also be sure to incorporate other reviewers' evaluation and comments and adjust the score if needed.

---

### Official Review · Reviewer_6Mzc · 2026-03-12

**Soundness:** 3
**Presentation:** 3
**Significance:** 3
**Originality:** 3
**Overall Recommendation:** 4
**Confidence:** 3

**Summary:**

The authors present a system for “automated AI research” that pursues and implements self-generated research ideas. The authors test this cycle using two GPU-intensive tasks and find that evolutionary search effectively uses this feedback to find solutions that outperform baseline values and even human experts. Reinforcement Learning (RL) appears to improve average but not top performance.

**Compliance With Llm Reviewing Policy:**

Affirmed.

**Final Justification:**

My initial concerns were addressed, and I appreciate the author's revision. I raise my score to Weak Accept.

**Key Questions For Authors:**

1) What can the authors say about transferring the powerful ideas discovered on a small scale to larger model sizes or other data sets?
2) Who can actually use this approach with these high computing costs? Is there a possibility for efficient scaling, or should the study be understood as a costly proof of concept?
3) Are there other possible applications to expand the scope of application?

**Limitations:**

Yes, in Appendix A.2.

**Strengths And Weaknesses:**

*Soundness strengths:*
- The system design is thorough, the experiments are clearly described, the metrics are well-defined with anti-reward-hacking measures, and the analyses (idea classification, thinking length dynamics, diversity collapse) are careful.
- The authors successfully built a robust executor capable of implementing and running hundreds of model-generated ideas in parallel, achieving >90% execution rates on challenging pre-training tasks.

*… weaknesses:*
- Error bars are missing. At least some bootstrapping would be helpful.
- The human expert comparison is asymmetric (a graduate class leaderboard for GRPO vs. a competitive speedrun leaderboard for nanoGPT where the gap to humans remains huge at 19.7 vs. 2.1 minutes).

*Presentation good, except:*
- The Discussion section would be better placed in the main texts, and not in the appendix as currently placed.

*Originality strengths:*
- The authors combine existing methods (evolutionary search and RL) for the automated implementation of ideas in realistic research problems. The decision to avoid simpler toy models appears to be original.
- This results in novel observations, such as the fact that longer thought processes are associated with lower execution rates. Future work could build on this.

*… weakness:*
- The relationship to AlphaEvolve is acknowledged but could the relation/differences be explained in more detail?

*Significance strengths:*
- Demonstrating the possibility of such a system, including RL.

*… weaknesses:*
- The experimental scope is narrow (only two environments, both in LLM training). No comparison against competing automated research systems.
- The RL results are, to some extend a negative finding of limited generality. It remains unclear (despite the attempts in A3), how general the RL finding is.
- The enormous compute cost (256–1024 GPUs per RL batch) goes largely unexamined.

---

> ### Author Rebuttal · Authors · 2026-03-31
>
> - *"Error bars are missing. At least some bootstrapping would be helpful."*
>
> We will add confidence intervals via bootstrapping for our evolutionary search experiments.
>
> - *"The human expert comparison is asymmetric (a graduate class leaderboard for GRPO vs. a competitive speedrun leaderboard for nanoGPT where the gap to humans remains huge at 19.7 vs. 2.1 minutes)."*
>
> We appreciate the reviewer for flagging this asymmetry. The choice of human baselines was primarily driven by data availability — the graduate class leaderboard was the most accessible benchmark for GRPO, while the competitive speedrun leaderboard was the natural reference point for nanoGPT.
>
> Rather than claiming parity with human researchers in either environment, the comparison is meant to provide a rough sense of scale — situating our system's outputs relative to what humans with domain knowledge can achieve. We have clarified the framing around these comparisons in the paper to make explicit that the two human baselines are not directly comparable (last paragraph in Sec 4.2), and we will add an explanation to the caption of Table 1 as well to clarify that the human comparison should be interpreted within each environment independently rather than across them, and that we do not make any claims of our system beating the best human experts.
>
> - *"The Discussion section would be better placed in the main texts, and not in the appendix as currently placed."*
>
> Agreed. We will move it to the main paper for the camera-ready version.
>
> - *"The relationship to AlphaEvolve is acknowledged but could the relation/differences be explained in more detail?"*
>
> On a high level, both AlphaEvolve and our evolutionary search method use LLMs in an evolutionary loop with automated execution feedback — but the two works target fundamentally different domains and regimes. AlphaEvolve targets tasks with fast and deterministic fitness functions where both the code implementation and solution evaluation are relatively straightforward (e.g., circle packing). In contrast, our work targets automated AI research, where the ideas involve implementing non-trivial algorithm interventions and running extensive GPU experiments for evaluation. Our work is the first to demonstrate that LLMs can implement a large fraction of these open-ended AI research ideas and can learn from the execution feedback to improve the idea effectiveness.
>
> Furthermore, we also investigate RL as an alternative learning algorithm to evolutionary search, a comparison entirely absent from AlphaEvolve's setting. More broadly, our focus is on understanding: (1) to what extent is automated execution of open-ended AI research ideas feasible; and (2) how well can LLMs learn from execution feedback via both evolutionary search and RL. These are important research questions that AlphaEvolve does not touch on at all. We will add a paragraph to the related work section making these distinctions more explicit.
>
> - *"The experimental scope is narrow (only two environments, both in LLM training)."*
>
> We appreciate this feedback and agree that evaluating across a broader range of research domains would strengthen our results. That said, we believe the two environments used in our paper do represent a meaningful and very non-trivial testbed: they involve complex open-ended search spaces and require careful code execution, and they represent two of the most important pieces of the modern LLM development stack. As far as we know, our two environments are among the most realistic and challenging research environments that the literature has worked on, where evaluating each idea involves extensive GPU experiments.
>
> Extending the framework to additional domains, such as architecture search beyond nanoGPT, data augmentation discovery, or broader ML systems and AI tasks, is an exciting direction, and we view it as a natural next step for future work. We do think our ideator-executor framework is relatively domain-agnostic — the primary requirement for transfer is that the idea generator has sufficient domain knowledge to propose valid and executable ideas and the idea executor can implement them correctly, which we expect to hold true, especially as frontier models and coding agents continue to advance.

---

> > ### Author Rebuttal · Reviewer_6Mzc · 2026-04-04
> >
> > Thank you for the response and the transparency and revision.

---

### Official Review · Reviewer_YLNR · 2026-03-13

**Soundness:** 3
**Presentation:** 3
**Significance:** 3
**Originality:** 3
**Overall Recommendation:** 4
**Confidence:** 4

**Summary:**

This paper studies execution-grounded automated AI research, motivated by the observation that LLMs can generate plausible ideas that often fail when executed. The authors build a high-throughput automated idea executor that turns natural-language research ideas into code diffs, runs large-scale GPU experiments in parallel, and returns execution-based performance as feedback. They instantiate two realistic, GPU-intensive research environments: LLM pre-training (a nanoGPT speedrun-style setting) and LLM post-training (improving a GRPO-based training setup). Using this infrastructure, the paper investigates whether LLM ideators can learn from execution feedback via two popular paradigms: (i) execution-guided evolutionary search (a scaffold combining exploration and exploitation across epochs) and (ii) reinforcement learning from execution reward (GRPO-style fine-tuning of an ideator model). Empirically, execution-guided search finds improvements over provided baselines within a small number of epochs and the paper provides extensive analyses of idea types, scaling behaviors across LLM backbones, and RL dynamics. A key finding is that while RL can improve average execution reward, it fails to improve the upper bound and tends to suffer from mode/diversity collapse, motivating future algorithmic work.

**Compliance With Llm Reviewing Policy:**

Affirmed.

**Final Justification:**

Some of my concerns are addressed by the authors during the rebuttal period. The remaining issues primarily relate to scope and generality rather than flaws in methodology or evaluation, and they are not easily resolved within a short rebuttal cycle, so I would like to maintain my original score.

**Key Questions For Authors:**

1. Will you release the executor code, environment definitions, and full execution logs/trajectories (including failed runs)? What parts are difficult to release (e.g., infra dependence), and what minimal artifacts can you provide so others can reproduce core results?
2. Can you report a clearer compute and wall-clock cost breakdown for (i) search epochs, (ii) best-of-N baselines, and (iii) RL training? This would help readers understand practical trade-offs and sample-efficiency claims.
3. How important are the exploitation/exploration split and annealing schedule? Do simpler alternatives (pure best-of-N, bandit-style selection, or different replay strategies) close the gap?
4. You show RL improves average reward but not max reward and leads to diversity collapse. Have you tried alternative RL algorithms or exploration-encouraging objectives (e.g., novelty bonuses, uncertainty-driven exploration, or curriculum over “idea difficulty”)? Which directions look most promising based on your analyses?
5. How do you expect the executor and the conclusions (e.g., search vs RL behaviors, scaling saturation) to transfer to other open-ended research domains beyond nanoGPT/GRPO?

**Limitations:**

Yes

**Strengths And Weaknesses:**

# Strengths:
- The paper tackles a hard-to-evaluate problem (open-ended research ideation) by grounding it in execution-based feedback with clearly defined environments, metrics, and automated evaluation pipelines.
- The proposed executor and experimental protocol enable large-scale, parallel execution of ideas and systematic comparisons across search/RL variants and LLM backbones.
- The empirical study is broad and includes meaningful analyses beyond headline numbers (e.g., search vs best-of-N under matched sampling budgets, stratifying ideas into hyperparameter vs algorithmic changes, and diagnosing RL collapse dynamics).
- While the core learning components (evolutionary search and GRPO-style RL) are not fundamentally new, their integration into a large-scale execution-grounded research loop, plus the construction of realistic LLM training environments and extensive empirical analyses, constitute a useful contribution.

# Weaknesses:
- Despite many experiments, the evaluation is still limited to two environments that are both LLM-training-centric. It remains unclear how well the executor and conclusions transfer to other open-ended research domains (e.g., architecture search beyond nanoGPT, data/augmentation discovery, or broader ML systems tasks).
- Some improvements appear driven by strong hyperparameter tuning and/or “known good tricks”; it would help to more clearly separate “rediscovering known recipes” from genuinely novel algorithmic insights, and to quantify stability across seeds and budget variations.

---

> ### Author Rebuttal · Authors · 2026-03-31
>
> We thank the reviewer for their thoughtful and constructive feedback. Given the scale of our experimental setup and the compute resources involved, adding new experiments within the rebuttal period is not feasible. However, we try our best to address some concerns below, and we hope this helps clarify any remaining questions.
>
> - *"Some improvements appear driven by strong hyperparameter tuning and/or “known good tricks”; it would help to more clearly separate “rediscovering known recipes” from genuinely novel algorithmic insights"*
>
> We partially address this concern via Table 2, where we classify generated ideas into hyper-parameter and algorithmic categories, showing that the majority involve algorithmic changes rather than simple hyper-parameter tuning, and these algorithmic ideas often lead to the best empirical performance. Regarding novelty detection, we note that we do not make novelty a claimed contribution of this work — our focus is specifically on whether LLMs can learn to generate more effective ideas that lead to more desired execution outcomes through execution grounding, which is an orthogonal question to whether those ideas are rediscoveries or genuinely novel. We agree that evaluating the novelty of LLM-generated ideas is an important and non-trivial open problem, and we flag it as a valuable direction for future work.
>
> - *"Will you release the executor code, environment definitions, and full execution logs/trajectories (including failed runs)? What parts are difficult to release (e.g., infra dependence), and what minimal artifacts can you provide so others can reproduce core results?"*
>
> Yes, we will release our environments and full trajectories from all of our evolutionary search and RL runs, and we will include the links in our camera-ready draft. The idea executor system includes scripts specific to our internal infra (e.g., GPU job allocation and submission), which we probably would not release. Nevertheless, it should be feasible for the community to reproduce and verify the experiment results of any LLM-generated ideas and codebases (these are included in our released trajectories) on our pretraining or posttraining environments.
>
> - *"Can you report a clearer compute and wall-clock cost breakdown for (i) search epochs, (ii) best-of-N baselines, and (iii) RL training? This would help readers understand practical trade-offs and sample-efficiency claims."*
>
> Every pretraining experiment takes ~25 minutes on 8 GPUs, and every post-training experiment takes ~45 minutes on 1 GPU. We use a batch size of 80 for pretraining experiments and a batch size of 50 for post-training experiments in evolutionary search, meaning each batch launches 80 pretraining jobs (80 x 8 GPUs) that run concurrently or 50 post-training jobs (50 x 1 GPUs) that run concurrently. After each batch is done, the next batch of ideas gets proposed and executed.
> The RL experiment is similar, except that we used a group size of 256 and 128 for post-training and pretraining, respectively.
>
> - *"Do simpler alternatives (pure best-of-N, bandit-style selection, or different replay strategies) close the gap?"*
>
> We compared our evolutionary search scaffold with a simple best-of-N baseline in Figure 4, where we show that our evolutionary search clearly dominates best-of-N under the same budget.
>
> - *"You show RL improves average reward but not max reward and leads to diversity collapse. Have you tried alternative RL algorithms or exploration-encouraging objectives (e.g., novelty bonuses, uncertainty-driven exploration, or curriculum over “idea difficulty”)? Which directions look most promising based on your analyses?"*
>
> We explored some of these directions in Appendix A.3. In particular, our experiments with a novelty/diversity bonus showed early promise in sustaining exploration and mitigating diversity collapse, though validating its effectiveness at scale remains an open question we leave for future work.
>
> On the average vs. max reward gap specifically, recent concurrent work has explored RL objectives that directly optimize for max-reward rollouts (e.g., "Learning to Discover at Test Time", arXiv 2026). We agree this is a promising direction, and investigating whether such objectives could improve RL performance in our environments is a natural next step we hope future work will explore.
>
> - *"Despite many experiments, the evaluation is still limited to two environments that are both LLM-training-centric. It remains unclear how well the executor and conclusions transfer to other open-ended research domains (e.g., architecture search beyond nanoGPT, data/augmentation discovery, or broader ML systems tasks)."*
>
> Due to space constraints, we refer you to the last point of our response to reviewer 6Mzc.

---

> > ### Author Rebuttal · Reviewer_YLNR · 2026-04-03
> >
> > The authors provide a clear and thoughtful rebuttal that addresses several of my concerns. I appreciate their detailed clarifications. Overall, the authors’ responses satisfactorily address most of my concerns within the constraints of the rebuttal period. The remaining issues primarily relate to scope and generality rather than flaws in methodology or evaluation, and they are not easily resolved within a short rebuttal cycle. I would like to maintain my original score. I remain supportive of acceptance, as the strengths of the work outweigh its limitations.

---

### Official Review · Reviewer_Dpwt · 2026-03-13

**Soundness:** 3
**Presentation:** 3
**Significance:** 4
**Originality:** 3
**Overall Recommendation:** 5
**Confidence:** 4

**Summary:**

The paper studies the AI research capabilities of current language by providing them access to an execution environment. The execution environment is a system that consists of an implementer which given an idea proposes a code diff for implementing it, a scheduler which allocates the compute to each experiment and a worker that executes the code for the idea. The paper considers two tasks from LLM research: nanoGPT pretraining and GRPO-based RL post-training. First, the authors evaluate frontier models as generators and implementers, and observe substantial number of executable ideas, with especially high completion rates under self-execution. Next, the authors explore using the execution environment as a source of rewards for guiding search via execution-guided evolutionary search and RL. The AlphaEvolve-style evolutionary search considered works surprisingly well and is more sample-efficient than best-of-N. By contrast, RL from execution reward improves the average quality of sampled ideas but not the best ideas, and suffers from a collapse in the diversity of the ideas generated.

**Compliance With Llm Reviewing Policy:**

Affirmed.

**Final Justification:**

My final recommendation is that the paper should be accepted for publication at ICML. The authors' rebuttal reinforced my positive assessment of the paper as a timely and well-structured exploration of LLM orchestrated and implement LLM research. The authors addressed my minor comments regarding the name.

**Key Questions For Authors:**

* In the RL training you find that the thinking length decreases consistently and there is a collapse in diversity. I wonder if this is due to the 8192 max length for the generator. That length seems quite short and it might be possible that the complex ideas that the model might want to generate are bottleneck by this maximum sequence length - which ultimately forces the model to get stuck at produce simpler ideas.
* How significantly do you expect the results to change if the whole generator-executor loop is incorporated into a single general purpose harness (e.g. Claude Code)?

**Limitations:**

Appendix A2 discusses the limitations of the work.

**Strengths And Weaknesses:**

Strengths:
* The paper is quite timely and pretty ambitious in terms of its scope and the problem that it tackles. As the capabilities of models grow rapidly, papers like this which push the the models to their limit provide important empirical verification of their abilities.
* The paper also goes beyond evaluating baseline agents over the models and includes more sophisticated evolutionary search / RL variants and highlight the strengths and weaknesses.
* Unlike most papers within the "AI Scientist" subfield, the paper is quite well written and communicates the results, both positive and negative, accurately without much overclaiming.
* The paper also provides a pretty good coverage of the most capable models (at the time of submission).

Weaknesses:
* While I really liked the ambition of the paper, I think the "Automated AI Research" in the title might be a bit too strong of a claim considering the empirical results are only on LLM based tasks. I suggest the authors modify it to "Automated LLM Research" to reflect the results more accurately. (I do understand that LLMs are quite popular in the field but it is, in my opinion, not ideal to equate all AI research with LLM research).
* Another major shortcoming is the lack of any AutoML baselines to contextualize the gains from the LLM guided ideas. I think at least some basic (compute-matched) hyperparameter optimization baselines would be useful to have. I understand that this quite a significant ask considering the scale of the experiments already performed, and may be hard to add due to the resources needed but would make the results much more informative.
* Another experiment which would be quite informative is running the evolutionary search with Qwen3-30B-A3B used for the RL policy. That would make for a nice comparison of what is more suited for this search setting, since as the authors themselves note, the goal in the discovery setting is the quality of the best sampled idea rather than the quality of the average idea. As such, that is a search problem whereas RL optimizes for the expected quality. So it seems to me that the evolutionary search setup should be more reliable for this setting.
* The setting considered here is a single diff update-style setting but a more realistic setting would be a multi-turn setting where the agent can perform experiments sequentially observing the results and behavior of each change before executing the next experiment.

---

> ### Author Rebuttal · Authors · 2026-03-31
>
> We thank the reviewer for their thoughtful and constructive feedback. Given the scale of our experimental setup and the compute resources involved, adding new experiments within the rebuttal period is not feasible. However, we try our best to address some concerns below, and we hope this helps clarify any remaining questions.
>
> - *"While I really liked the ambition of the paper, I think the "Automated AI Research" in the title might be a bit too strong of a claim considering the empirical results are only on LLM based tasks. I suggest the authors modify it to "Automated LLM Research" to reflect the results more accurately. (I do understand that LLMs are quite popular in the field but it is, in my opinion, not ideal to equate all AI research with LLM research)."*
>
> Thank you for this suggestion. We appreciate the reviewer's attention to precision in scope. While we think our approach is generally applicable to a wide range of empirical AI problems beyond just LLM research, we do agree our experiments in this current draft are specifically targeted at LLM research, and we are happy to update the title to "Automated LLM Research" to more accurately reflect the empirical scope of this work.
>
> - *"Another major shortcoming is the lack of any AutoML baselines to contextualize the gains from the LLM guided ideas. I think at least some basic (compute-matched) hyperparameter optimization baselines would be useful to have. I understand that this quite a significant ask considering the scale of the experiments already performed, and may be hard to add due to the resources needed but would make the results much more informative."*
>
> Yes, we agree this would be a valuable comparison, but we also think that our existing results are themselves very worthwhile since: (a) as shown in Table 2 of our paper, our approach allows the generation of many algorithmic ideas beyond just hyper-parameter tuning, and these algorithmic ideas often achieve the best empirical performance. This sets us apart from the baselines that only tune the hyper-parameters; and (b) we have careful studies on how execution grounding affects LLM-based autoresearch, which is a result of its own interest due to how much people are investing in this direction.
>
> - *"In the RL training you find that the thinking length decreases consistently and there is a collapse in diversity. I wonder if this is due to the 8192 max length for the generator. That length seems quite short and it might be possible that the complex ideas that the model might want to generate are bottleneck by this maximum sequence length - which ultimately forces the model to get stuck at produce simpler ideas."*
>
> We analyzed the reasoning traces and ideas from our RL runs and observed no truncation (the longest rollout contains a thinking trace of 7311 tokens and an idea of 428 tokens, below our max seq length limit). We thus rule out this hypothesis and conclude that the probable explanation is that ideas with longer thinking traces tend to be more complicated to execute, leading the model to favor simpler ideas instead, since failed execution would result in zero reward (as shown in Figure 6).
>
> - *"How significantly do you expect the results to change if the whole generator-executor loop is incorporated into a single general-purpose harness (e.g., Claude Code)?"*
>
> While a controlled comparison is outside the scope of the current work, our intuition is that execution rates would likely be higher with Claude Code, given its substantially larger action space compared to our scaffold (which uses repeated sampling with sequential revision of code diffs, as described in Sec 2.2). It is also possible that Claude Code could autonomously discover appropriate evolution strategies — balancing exploration and exploitation — without explicit scaffolding.
>
> That said, we believe a carefully designed scaffold offers meaningful advantages in controllability and scalability that a general-purpose harness may not easily replicate. More broadly, how an expert-designed evolution scaffold compares against a fully autonomous system like Claude Code (in the spirit of Andrej Karpathy's autoresearch) is a genuinely interesting open question, and one we hope future work will address systematically.

---

> > ### Author Rebuttal · Reviewer_Dpwt · 2026-04-02
> >
> > Thanks for the responses! I maintain my positive score for the paper.

---

### Decision · Program_Chairs · 2026-04-30

**Decision:**

Accept (regular)

**Comment:**

The paper introduces an automated execution framework for LLM research, evaluating how models learn from execution feedback through evolutionary search and reinforcement learning. Reviewers recognize the work as timely and technically ambitious, specifically praising the design of the automated executor and the insightful analysis regarding diversity collapse and the discrepancy between average and max rewards in RL. During the rebuttal, the authors successfully addressed concerns regarding the title's scope by narrowing it to LLM research and provided necessary clarifications on compute costs and experimental baselines. The submission is technically sound, well-written, and offers significant empirical value to the community. Therefore, the paper is recommended for acceptance.